# Anonymous and Copy-Robust Delegations for Liquid Democracy[*]

**Markus Utke**
Department of Mathematics and Computer Science
TU Eindhoven
Eindhoven, The Netherlands
m.utke@tue.nl

**Ulrike Schmidt-Kraepelin**
Simons Laufer Mathematical Sciences
Institute (SLMath)
Berkeley, CA, United States
uschmidt@slmath.org

## Abstract

Liquid democracy with ranked delegations is a novel voting scheme that unites the practicability of representative democracy with the idealistic appeal of direct democracy: Every voter decides between *casting* their vote on a question at hand or *delegating* their voting weight to some other, trusted agent. Delegations are transitive, and since voters may end up in a delegation cycle, they are encouraged to indicate not only a single delegate, but a set of potential delegates and a ranking among them. Based on the delegation preferences of all voters, a *delegation rule* selects one representative per voter. Previous work has revealed a trade-off between two properties of delegation rules called *anonymity* and *copy-robustness*.

To overcome this issue we study two *fractional* delegation rules: MIXED BORDA BRANCHING, which generalizes a rule satisfying copy-robustness, and the RANDOM WALK RULE, which satisfies anonymity. Using the *Markov chain tree theorem*, we show that the two rules are in fact equivalent, and simultaneously satisfy generalized versions of the two properties. Combining the same theorem with *Fulkerson's algorithm*, we develop a polynomial-time algorithm for computing the outcome of the studied delegation rule. This algorithm is of independent interest, having applications in semi-supervised learning and graph theory.

## 1 Introduction

Today, democratic decision-making in legislative bodies, parties, and non-profit organizations is often done via one of two extremes: In *representative democracy*, the constituents elect representatives who are responsible for deciding upon all upcoming issues for a period of several years. In *direct democracy*, the voters may vote upon every issue themselves. While the latter is distinguished by its idealistic character, it may suffer from low voter turnout as voters do not feel sufficiently informed. Liquid democracy aims to provide the best of both worlds by letting voters decide whether they want to *cast* their opinion on an issue at hand, or prefer to *delegate* their voting weight to some other, trusted voter. Delegations are transitive, i.e., if voter $v_1$ delegates to voter $v_2$, and $v_2$ in turn delegates to voter $v_3$, who casts its vote, then $v_3$ receives the voting weight of both $v_1$ and $v_2$. Liquid democracy has been implemented, for example, by political parties [Kling et al., 2015] and Google [Hardt and Lopes, 2015]. From a theoretic viewpoint, liquid democracy has been studied intensively by the social choice community in the last decade [Paulin, 2020].

Earlier works on liquid democracy [Christoff and Grossi, 2017, Brill, 2018] have pointed towards the issue of *delegation cycles*, e.g., the situation that occurs when voter $v_3$ in the above example decides to delegate to voter $v_1$ instead of casting its vote. If this happens, none of the three voters reaches a

---

[*]Part of this research was carried out while both authors were affiliated with TU Berlin and Universidad de Chile, and Markus Utke was affiliated with University of Amsterdam.

37th Conference on Neural Information Processing Systems (NeurIPS 2023).

casting voter via a chain of trusted delegations, and therefore their voting weight would be lost. In order to reduce the risk of the appearance of such so-called *isolated* voters, several scholars suggested to allow voters to indicate *back-up* delegations [Brill, 2018, Gölz et al., 2021, Kavitha et al., 2022] that may be used in case there is no delegation chain using only top-choice delegations. In *liquid democracy with ranked delegations* [Brill et al., 2022, Colley et al., 2022, Kavitha et al., 2022], voters are assumed to indicate a set of trusted delegates together with a ranking (preference order) among them. In fact, Brill et al. [2022] showed empirically that in many random graph models, one to two back-up delegations per voter suffice in order to avoid the existence of isolated voters almost entirely.

Allowing the voters to indicate multiple possible delegations calls for a principled way to decide between multiple possible delegation chains. For example, consider Figure 1: Should voter $v_1$'s weight be assigned to casting voter $s_1$, via $v_1$'s second-ranked delegation, or should it rather be assigned to voter $s_2$ by following $v_1$'s first-ranked delegation to voter $v_2$, and then following $v_2$'s second-ranked delegation? Brill et al. [2022] introduced the concept of *delegation rules*, which take as input a *delegation graph* (i.e., a digraph with a rank function on the edges) and output an assignment of each (non-isolated) delegating voter to a casting voter. In order to navigate within the space of delegation rules, they apply the axiomatic method [Thomson, 2001], as commonly used in social choice theory. In particular, the authors argue that the following three axioms are desirable:

(i) *Confluence*: A delegation rule selects one path from every delegating voter to a casting voter and these paths are "consistent" with one another. That is, when the path of voter $v_1$ reaches some other delegating voter $v_2$, the remaining subpath of $v_1$ coincides with the path of $v_2$. This was argued to increase accountability of delegates [Brill et al., 2022, Gölz et al., 2021].

(ii) *Anonymity*: A delegation rule should make decisions solely on the structure of the graph, not on the identities of the voters, i.e. it should be invariant under renaming of the voters.

(iii) *Copy-robustness*: If a delegating voter $v_1$ decides to cast her vote herself instead of delegating, this should not change the sum of the voting weight assigned to $v_1$'s representative and herself. This property was emphasized by practitioners [Behrens and Swierczek, 2015] to avoid manipulations in the system by delegating voters acting as casting voters but actually *copying* the vote of their former representative.

Figure 1: Example of a delegation graph. Delegating voters ($v_1$ and $v_2$) are indicated by circles and casting voters ($s_1$ and $s_2$) by squares. Solid edges represent the first-ranked delegations and dashed edges second-ranked delegations.

For any pair of axioms (i) to (iii), Brill et al. [2022] provide a delegation rule that satisfies both of them. In contrast, we prove in Section 7 that there exists no delegation rule that satisfies all three properties simultaneously, thereby strengthening an impossibility result by Brill et al. [2022].[2]

**Our Contribution**   We show that the above impossibility is due to the restriction that delegation rules may not distribute the voting weight of a delegating voter to more than one casting voter.

- We generalize the definition of delegation rules to *fractional delegation rules* (Section 3) and provide generalizations of all three axioms above (Section 7).

- We introduce a natural variant of the BORDA BRANCHING rule [Brill et al., 2022], which we call MIXED BORDA BRANCHING. We show that this rule is equivalent to the RANDOM WALK RULE, a fractional delegation rule that has been suggested by Brill [2018].

- In our main result, we build upon *Fulkerson's algorithm* [Fulkerson, 1974] and the *Markov chain tree theorem* [Leighton and Rivest, 1986] and show the existence of a polynomial-time algorithm for MIXED BORDA BRANCHING. This algorithm is of independent interest, as it computes the probability of two nodes being connected, when sampling a min-cost branching in a digraph uniformly at random. This problem features in semi-supervised learning, under the name *directed power watershed* [Fita Sanmartin et al., 2021], a directed

---

[2]Brill et al. [2022] show that there exists no delegation rule belonging to the subclass of *sequence rules* that is both confluent and copy-robust. Any sequence rule is in particular anonymous.

variant of the *power watershed* [Couprie et al., 2010]. To the best of our knowledge, we provide the first efficient algorithm.

- In Section 7, we show that the RANDOM WALK RULE (and thus MIXED BORDA BRANCH-ING) satisfies the generalizations of all three axioms. We also formalize the impossibility for non-fractional delegation rules. Beyond that, we show that the RANDOM WALK RULE satisfies a generalization of a further axiom (*guru participation*) which has been studied in the literature [Kotsialou and Riley, 2020, Colley et al., 2022, Brill et al., 2022] (Appendix C).

The proofs (or their completions) for results marked by (★) can be found in the appendix.

**Related Work**    *Liquid democracy.* The idea to let agents rank potential delegates in liquid democracy was first presented by the developers of the liquid democracy platform *Liquid Feedback* [Behrens and Swierczek, 2015], who presented seven properties that cannot be satisfied simultaneously. Some of these properties, such as *copy-robustness* and *guru participation*, have been picked up in the social choice literature [Kotsialou and Riley, 2020, Brill et al., 2022, Colley et al., 2022]. The connection of confluent delegation rules to branchings in a digraph was first emphasized by Kavitha et al. [2022] and later built upon in [Brill et al., 2022, Natsui and Takazawa, 2022]. We base our model on [Brill et al., 2022], as their model captures all rules and axioms from the literature. Fractional delegations were studied by Degrave [2014] and Bersetche [2022], however, here agents indicate a desired distribution among their delegates instead of a ranking. While the two approaches are orthogonal, we argue in Appendix B that they could be easily combined (and our algorithm could be adjusted).

*Branchings and matrix tree theorems.* Our algorithm for computing MIXED BORDA BRANCHING is based on an algorithm for computing min-cost branchings in directed trees. This can be done, e.g., via Edmond's algorithm [Edmonds, 1967] or Fulkerson's algorithm [Fulkerson, 1974]. The latter comes with a characterization of min-cost branchings in terms of dual certificates, which we utilize in Algorithm 2. We refer to Kamiyama [2014] for a comprehensive overview on the literature of min-cost branchings. Our algorithm makes use of (a directed version of) the *matrix tree theorem* [Tutte, 1948], which allows to count directed trees in a digraph. An extension of this theorem is the *Markov chain tree theorem* [Leighton and Rivest, 1986], which we use for the construction of our algorithm as well as for proving the equivalence of MIXED BORDA BRANCHING and the RANDOM WALK RULE. A comprehensive overview of the literature is given by Pitman and Tang [2018].

*Semi-supervised learning.* There is a connection of our setting to graph-based semi-supervised learning. In particular, Algorithm 2 is related to the *power watershed algorithm* [Couprie et al., 2010] and the *probabilistic watershed algorithm* [Fita Sanmartin et al., 2019, 2021]. We elaborate on this connection in Section 4.

## 2   Preliminaries

The main mathematical concepts used in this paper are directed graphs (also called digraphs), branchings, in-trees, and Markov chains, all of which we briefly introduce below.
We assume that a digraph $G$ has no parallel edges, and denote by $V(G)$ the set of nodes and by $E(G)$ the set of edges of $G$. We use $\delta_G^+(v)$ to indicate the set of outgoing edges of node $v \in V(G)$, i.e., $\delta_G^+(v) = \{(v, u) \in E(G)\}$. For a set of nodes $U \subseteq V(G)$, we define the outgoing cut of $U$ by $\delta_G^+(U) = \{(u, v) \in E(G) \mid u \in U, v \in V(G) \setminus U\}$. A walk $W$ is a node sequence $(W_1, \ldots, W_{|W|})$, such that $(W_i, W_{i+1}) \in E(G)$ for $i \in \{1, \ldots, |W| - 1\}$. We omit $G$ if it is clear from the context.

**Branchings and in-trees.** Given a digraph $G$, we say that $B \subseteq E$ is a branching (or in-forest) in $G$, if $B$ is acyclic, and the out-degree of any node $v \in V(G)$ in $B$ is at most one, i.e., $|B \cap \delta^+(v)| \leq 1$. Throughout the paper, we use the term *branching* to refer to *maximum-cardinality branchings*, i.e., branchings $B$ maximizing $|B|$ among all branchings in $G$. For a given digraph $G$ we define $\mathcal{B}(G)$ as the set of all (max-cardinality) branchings and $\mathcal{B}_{v,s}(G)$ as the set of all (max-cardinality) branchings in which node $v \in V(G)$ has a path to the node $s \in V(G)$. For any branching $B$ in any digraph $G$ it holds that $|B| \leq |V(G)| - 1$. If in fact $|B| = |V(G)| - 1$, then $B$ is also called an in-tree. For every in-tree $T \subseteq E$, there exists exactly one node $v \in V(G)$ without outgoing edge. In this case, we also say that $v$ is the sink of $T$ and call $T$ a $v$-tree. For $v \in V(G)$, we let $\mathcal{T}_v(G)$ be the set of $v$-trees in $G$.

**Matrix tree theorem.** For our main result we need to count the number of in-trees in a weighted digraph, which can be done with help of the matrix tree theorem. For a digraph $G$ with weight

function $w : E \to \mathbb{N}$, we define the weight of a subgraph $T \subseteq E$ as $w(T) = \prod_{e \in T} w(e)$ and the weight of a collection of subgraphs $\mathcal{T}$ as $w(\mathcal{T}) = \sum_{T \in \mathcal{T}} w(T)$. Then, we define the *Laplacian* matrix of $(G, w)$ as $L = D - A$, where $D$ is a diagonal matrix containing the weighted out-degree of any node $v$ in the corresponding entry $D_{v,v}$ and $A$ is the weighted adjacency matrix, given as $A_{u,v} = w((u,v))$ for any edge $(u,v) \in E$ and zero everywhere else. Moreover, we denote by $L^{(v)}$ the matrix resulting from $L$ when deleting the row and column corresponding to $v$.

**Lemma 1** (Matrix tree theorem [Tutte, 1948, De Leenheer, 2020][3] ). *Let $(G, w)$ be a weighted digraph and let $L$ be its Laplacian matrix. Then,*

$$det(L^{(v)}) = \sum_{T \in \mathcal{T}_v} \prod_{e \in T} w(e) = w(\mathcal{T}_v).$$

If we interpret the weight of an edge as its *multiplicity* in a multigraph, then $det(L^{(v)})$ equals the total number of distinct $v$-trees.

**Markov Chains.** A Markov chain is a tuple $(G, P)$, where $G$ is a digraph and the matrix $P \in [0,1]^{|V| \times |V|}$ encodes the transition probabilities. That is, the entry $P_{u,v}$ indicates the probability with which the Markov chain moves from state $u$ to state $v$ in one timestep. For a given edge $e = (u,v) \in E$, we also write $P_e$ to refer to $P_{u,v}$. For all $v \in V$ it holds that $\sum_{e \in \delta^+(v)} P_e = 1$. Moreover, if $(u,v) \notin E$, we assume $P_{u,v} = 0$. We define the matrix $Q \in [0,1]^{|V| \times |V|}$ as $Q = \lim_{\tau \to \infty} \frac{1}{\tau} \sum_{i=0}^{\tau} P^\tau$. If $(G, P)$ is an absorbing Markov chain, $Q_{u,v}$ corresponds to the probability for a random walk starting in $u$ to end in absorbing state $v$. In contrast, if $G$ is strongly connected and $P$ is positive for all edges in $G$, then $Q_{u,v}$ corresponds to the relative number of times $v$ is visited in an infinite random walk independent of the starting state [Grinstead and Snell, 1997].

## 3 Liquid Democracy with Fractional Delegations

A *delegation graph* $G = (N \cup S, E)$ is a digraph with a *cost function*[4] $c : E \to \mathbb{N}$ (called *rank function* before), representing the preferences of the voters over their potential delegates (lower numbers are preferred). Nodes correspond to voters and an edge $(u,v) \in E$ indicates that $u$ accepts $v$ as a delegate. By convention, the set of nodes $S$ corresponds exactly to the sinks of the digraph, i.e., the set of nodes without outgoing edges. Thus, $S$ captures the casting voters, and $N$ the delegating voters. We assume for all $v \in N$ that they reach some element in $S$.[5] A *delegation rule* maps any delegation graph to a *fractional assignment*, i.e., a matrix $A \in [0,1]^{N \times S}$, where, for every $v \in N$, $s \in S$, the entry $A_{v,s}$ indicates the fraction of delegating voter $v$'s weight that is allocated to casting voter $s \in S$.[6] For any $v \in N$ we refer to any casting voter $s \in S$ with $A_{v,s} > 0$ as a *representative* of $v$. For assignment $A$, the *voting weight* of a casting voter $s \in S$ is defined as $\pi_s(A) = 1 + \sum_{v \in N} A_{v,s}$. A *non-fractional* delegation rule is a special case of a delegation rule, that always returns assignments $A \in \{0,1\}^{N \times S}$.

**MIXED BORDA BRANCHING.** The output of any non-fractional, confluent delegation rule can be represented as a branching: Any branching in the delegation graph consists of $|N|$ edges, and every delegating voter has a unique path to some casting voter. Brill et al. [2022] suggested to select min-cost branchings, i.e., those minimizing $\sum_{e \in B} c(e)$. The authors call these objects Borda branchings and show that selecting them yields a copy-robust rule. As this rule is inherently non-anonymous, we suggest to mix uniformly over all Borda branchings, hoping to gain anonymity without losing the other properties[7]. Formally, for a given delegation graph $(G, c)$, let $\mathcal{B}^*(G)$ be the set of all Borda branchings and let $\mathcal{B}^*_{v,s}(G)$ be the set of all Borda branchings in which delegating voter $v \in N$ is connected to casting voter $s \in S$. MIXED BORDA BRANCHING returns the assignment $A$ defined as

$$A_{v,s} = \frac{|\mathcal{B}^*_{v,s}(G)|}{|\mathcal{B}^*(G)|} \quad \text{for all } v \in V, s \in S.$$

---

[3]Tutte [1948] proves the theorem for digraphs, the weighted version can be found in [De Leenheer, 2020].

[4]We remark that in this paper cost functions and weight functions play different roles.

[5]If this is not the case, such a voter is *isolated*, i.e., there is no chain of trusted delegations to a casting voter, thus we cannot meaningfully assign its voting weight. We remove isolated nodes in a preprocessing step.

[6]While this definition allows $A_{v,s} > 0$ for any $v \in N, s \in S$, for a sensible delegation rule, this should only be the case if there exists a path from $v$ to $s$. *Confluence* (defined in Section 7) implies this restriction.

[7]Confluence and copy-robustness are not directly inherited from the non-fractional counterpart of the rule.

**RANDOM WALK RULE.** The second delegation rule was suggested in Brill [2018] and attributed to Vincent Conitzer. For any given delegation graph $(G, c)$ and fixed $\varepsilon \in (0, 1]$, we define a Markov chain on $G$, where the transition probability matrix $P^{(\varepsilon)} \in [0, 1]^{V(G) \times V(G)}$ is defined as

$$P_{u,v}^{(\varepsilon)} = \frac{\varepsilon^{c(u,v)}}{\bar{\varepsilon}_u} \quad \text{for any } (u, v) \in E, \tag{1}$$

where $\bar{\varepsilon}_u = \sum_{(u,v) \in \delta^+(u)} \varepsilon^{c(u,v)}$ is the natural normalization factor. The Markov chain $(G, P^{(\varepsilon)})$ is absorbing for every $\varepsilon \in (0, 1]$ where the absorbing states are exactly $S$. The RANDOM WALK RULE returns the fractional assignment $A$ defined as the limit of the absorbing probabilities, i.e.,

$$A_{v,s} = \lim_{\varepsilon \to 0} \left( \lim_{\tau \to \infty} \frac{1}{\tau} \sum_{i=0}^{\tau} (P^{(\varepsilon)})^\tau \right)_{v,s} \quad \text{for all } v \in N, s \in S.$$

## 4    Connection to Semi-Supervised Learning

In graph-based semi-supervised learning, the input is a directed or undirected graph, where nodes correspond to data points and edges correspond to relationships between these. A subset of the nodes is *labeled* and their labels are used to classify unlabeled data. Many algorithms compute a fractional assignment of unlabeled data points towards labeled data points, which is then used to determine the predictions for unlabeled data. In the directed case, there exists a one-to-one correspondence to our model: Delegating voters correspond to unlabeled data and casting voters correspond to labeled data.

**Power Watershed.** In the undirected case, the *power watershed* [Couprie et al., 2010] can be interpreted as an undirected analog of MIXED BORDA BRANCHING: For any pair of unlabeled data point $x$ and labeled data point $y$, the algorithm computes the fraction of min-cost undirected maximum forests that connect $x$ to $y$.[8] The authors provide a polynomial-time algorithm for computing its outcome. On a high level, our algorithm (Section 5) is reminiscent of theirs, i.e., both algorithms iteratively contract parts of the graph, leading to a hierarchy of subsets of the nodes. However, while the algorithm by Couprie et al. [2010] only needs to carry out computations at the upper level of the hierarchy, our algorithm has to carry out computations at each level of the hierarchy. We believe that the increased complexity of our algorithm is inherent to our problem and do not find this surprising: Even the classic min-cost spanning tree problem can be solved by a greedy algorithm and forms a Matroid, but this structure gets lost when moving to its directed variant.

**Directed Probabilistic Watershed.** Fita Sanmartin et al. [2021] study a directed version of graph-based semi-supervised learning. The authors introduce the *directed probabilistic watershed (DProbWS)*: Similar to our model, there exists a cost function $c$ and a weight function $w$ on the edges. In their case, these functions are linked by $w(e) = \exp(-\mu c(e))$. The paper studies a Gibbs distribution with respect to the weights, i.e., the probability of sampling a branching is *proportional* to its weight. The parameter $\mu$ controls the entropy of this function, i.e., for $\mu = 0$, any branching is sampled with equal probability, while larger $\mu$ places more probability on low cost branchings. The authors show: (i) For fixed $\mu$, the fractional allocation induced by the Gibbs distribution can be computed by calculating the absorbing probabilities of a Markov chain. (ii) For the limit case $\mu \to \infty$, the defined distribution equals the uniform distribution over all min-cost branchings, i.e., the distribution that we study in this paper. In this limit case, the authors refer to the corresponding solution as the *directed power watershed*. Hence, the authors have shown the equivalence of the directed power watershed and the limit of a parameterized Markov chain. Since this Markov chain only slightly differs from ours, this result is very close to our Theorem 5. As its proof is very short and might be of interest for the reader, we still present it in Section 6. Importantly, Fita Sanmartin et al. [2021] do not show how to *compute* the outcome of the directed power watershed. In particular, the algorithm from (i) makes explicit use of the weight function on the edges (which depends on $\mu$). Hence, the running time of the algorithm grows to infinity in the limit case. We fill this gap by providing the (to the best of our knowledge) first polynomial-time algorithm for the directed power watershed. Maybe surprisingly, we need a significantly more complex approach to solve the limit case: While the algorithm of Fita Sanmartin et al. [2021] solves one absorbing Markov chain, our algorithm derives a hierarchical structure of subsets of the nodes by the structural insights provided by *Fulkerson's* algorithm, and solves several Markov chains for each of the hierarchy.

---

[8]This analogy holds for the variant of the power watershed when a parameter $q$ is 2 [Couprie et al., 2010].

# 5 Computation of MIXED BORDA BRANCHING

Our algorithm for computing the outcome of MIXED BORDA BRANCHING is an extension of an algorithm by Fulkerson [1974] for computing an arbitrary min-cost branchings in a digraph. The algorithm by Fulkerson follows a primal-dual approach and can be divided into two phases, where the first phase characterizes min-cost branchings with the help of a family of subsets of the nodes, and the second phase then constructs one arbitrary min-cost branching. Building upon the first phase, we show that, for every two nodes $v \in N$ and $s \in S$, we can *count* the number of min-cost branchings that connect the two nodes by applying an extension of the matrix tree theorem [Tutte, 1948, De Leenheer, 2020] and the *Markov chain tree theorem* [Leighton and Rivest, 1986].

**Fulkerson's algorithm.**[9] The algorithm (described formally in Algorithm 1) maintains a function $y : 2^{N \cup S} \to \mathbb{N}$ and a subset of the edges $E_y \subseteq E$. The set $E_y$ captures edges that are *tight* (w.r.t. $y$), i.e., those $e \in E$ satisfying

$$\sum_{X \subseteq N \cup S : e \in \delta^+(X)} y(X) = c(e).$$

The algorithm takes as input a delegation graph $G$ together with a cost function $c : E \to \mathbb{N}$.

---

**Algorithm 1** Fulkerson's Algorithm [Fulkerson, 1974, Kamiyama, 2014]

---

1: Set $y(X) = 0$ for any set $X \subseteq N \cup S$ except $y(\{s\}) = 1$ for any $s \in S$, $E_y = \emptyset$
2: **while** some node in $N$ cannot reach any node in $S$ in the graph $(N \cup S, E_y)$ **do**
3:     let $X \subseteq N$ be a strongly connected component in $(N \cup S, E_y)$ with $\delta^+(X) \cap E_y = \emptyset$
4:     set $y(X)$ to minimum value such that some edge in $\delta^+(X)$ is tight, add tight edges to $E_y$
5: $\mathcal{F} = \{X \subseteq N \cup S \mid y(X) > 0\}$
6: **return** $(\mathcal{F}, E_y, y)$

---

Since $c(e) \geq 1$ for all $e \in E$, the output $\mathcal{F}$ contains all singleton sets induced by nodes in $N \cup S$, and beyond that subsets of $N$. In the following we show several structural insights that are crucial for the construction of our algorithm. While statements (ii) and (iii) have been proven in similar forms by Fulkerson [1974], we prove all of Lemma 2 in Appendix A for completeness.

**Lemma 2 (★).** *Let $(G, c)$ be a delegation graph and let $(\mathcal{F}, E_y, y)$ be the output of Algorithm 1. Then:*

  *(i) For every $(G, c)$, the output of the algorithm is unique, i.e., it does not depend on the choice of the strongly connected component in line 3.*

  *(ii) $\mathcal{F}$ is laminar, i.e., for any $X, Y \in \mathcal{F}$ it holds that either $X \subseteq Y$, $Y \subseteq X$, or $X \cap Y = \emptyset$.*

  *(iii) Branching $B$ in $(G, c)$ is min-cost iff (a) $B \subseteq E_y$, and (b) $|B \cap \delta^+(X)| = 1$ for all $X \in \mathcal{F}, X \subseteq N$.*

  *(iv) For every $X \in \mathcal{F}$, an in-tree $T$ in $G[X] = (X, E[X])$, where $E[X] = \{(u, v) \in E \mid u, v \in X\}$, is min-cost iff (a) $T \subseteq E_y$, and (b) $|T \cap \delta^+(Y)| = 1$ for all $Y \in \mathcal{F}$ such that $Y \subset X$.*

**Intuition and notation for Algorithm 2.** For our algorithm, the crucial statements in Lemma 2 are (ii) and (iii). First, because $\mathcal{F}$ forms a laminar family, there exists a natural tree-like structure among the sets. We say that a set $Y \in \mathcal{F}$ is a *child* of a set $X \in \mathcal{F}$, if $Y \subset X$ and there does not exist a $Z \in \mathcal{F}$, such that $Y \subset Z \subset X$. Moreover, for some $X \in \mathcal{F}$ or $X = N \cup S$, we define $G_X = (V_X, E_X)$ as the tight subgraph that is restricted to the node set $X$ and contracts all children of $X$. Formally, $V_X = \{Y \mid Y \text{ is a child of } X\}$ and $E_X = \{(Y, Y') \mid (u, v) \in E_y, u \in Y, v \in Y', \text{ and } Y, Y' \in V_X\}$. In the following we first focus on the graph corresponding to the uppermost layer of the hierarchy, i.e., $G_X$ for $X = N \cup S$. Now, statement (iii) implies that every min-cost branching $B$ in $G$ leaves every child of $X$ exactly once and only uses tight edges. Hence, if we project $B$ to an edge set $\hat{B}$ in the contracted graph $G_X$, then $\hat{B}$ forms a branching in $G_X$. However, there may exist many min-cost branchings in $G$ that map to the same branching in $G_X$. The crucial insight is that we can construct a weight function $w_X$ on the edges of $G_X$, such that the weighted number of branchings in $G_X$ equals the number of min-cost branchings in $G$. This function is constructed by calculating for every child

---

[9]We slightly adjust the algorithm, as Fulkerson [1974] studies directed out-trees and assumes one sink only.

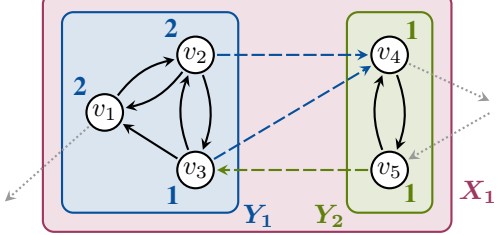
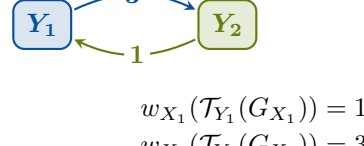

(a) The delegation graph $(G, c)$ restricted to the set $X_1$. Nodes $v \in Y_1$ are labeled with $t_{Y_1}(v)$ and nodes in $v \in Y_2$ are labeled with $t_{Y_2}(v)$. Recall, that $t_{Y_1}(v)$ is the total number of min-cost $v$-trees in $G[Y_1]$.

(b) Digraph $G_{X_1}$ with edge weights $w_{X_1}$. Based on this input, for $i \in \{1, 2\}$, we calculate the total weight of all $Y_i$-trees in $G_{X_1}$, i.e., $w_{X_1}(\mathcal{T}_{Y_i}(G_{X_1}))$.

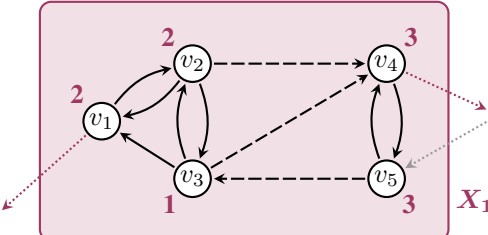
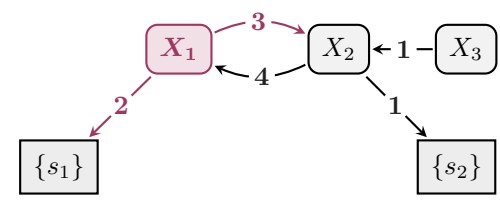

(c) The graph $(G, c)$ restricted to $X_1$. Every node $v \in X_1$ is labeled with $t_{X_1}(v)$, which was calculated by multiplying the weights from Figure 2a with the results from Figure 2b.

(d) An example for $(G_X, w_X)$, where $X = N \cup S$. In the last iteration, this graph is translated into a Markov chain and the absorbing probabilities are returned.

Figure 2: Two iterations of Algorithm 2. Costs are depicted by edge patterns (solid equals cost 1, dashed equals cost 2, and dotted equals cost 3) and weights are depicted by numbers on the edges.

$Y$ of $X$ and every node $v \in Y$, the number of min-cost $v$-trees inside the graph $G[Y] = (Y, E[Y])$, where $E[Y] = \{(u, v) \in E \mid u, v \in Y\}$. This number, denoted by $t_Y(v)$, can be computed by recursively applying the matrix tree theorem. Coming back to the graph $G_X$, however, we need a more powerful tool since we need to compute the (relative) number of weighted branchings in $G_X$ connecting any node to a sink node. Thus, we introduce the Markov chain tree theorem (in a slightly modified version so that it can deal with Markov chains induced by weighted digraphs).

For a weighted digraph $(G, w)$ we define the corresponding Markov chain $(G', P)$ as follows: The digraph $G'$ is derived from $G$ by adding self-loops, and for $(u, v) \in E(G')$ we let

$$
P_{u,v} = \begin{cases} \frac{w(u,v)}{\Delta} & \text{if } u \neq v \\ 1 - \frac{\sum_{e \in \delta^+(u)} w(e)}{\Delta} & \text{if } u = v, \end{cases}
$$

where $\Delta = \max_{v \in V} \sum_{e \in \delta^+(v)} w(e)$.

**Lemma 3** (Markov chain tree theorem (Leighton and Rivest [1986])). *Consider a weighted digraph $(G, w)$ and the corresponding Markov chain $(G', P)$ and let $Q = \lim_{\tau \to \infty} \frac{1}{\tau} \sum_{i=0}^{\tau} P^\tau$. Then, the entries of the matrix $Q$ are given by*

$$
Q_{u,v} = \frac{\sum_{B \in \mathcal{B}_{u,v}} \prod_{e \in B} P_e}{\sum_{B \in \mathcal{B}} \prod_{e \in B} P_e} = \frac{\sum_{B \in \mathcal{B}_{u,v}} \prod_{e \in B} w(e)}{\sum_{B \in \mathcal{B}} \prod_{e \in B} w(e)} = \frac{\sum_{B \in \mathcal{B}_{u,v}} w(B)}{\sum_{B \in \mathcal{B}} w(B)}.
$$

We formalize Algorithm 2, which takes as input a delegation graph $(G, c)$ and, in contrast to the intuition above, works in a bottom-up fashion. We refer to Figure 2 for an illustration.

**Theorem 4 (★).** *Algorithm 2 returns* MIXED BORDA BRANCHING *and runs in* poly$(n)$.

*Proof sketch.* The main part of the proof, shows by induction over the laminar hierarchy of $\mathcal{F}$, that for every $X \in \mathcal{F}$ and $v \in X$, the value $t_X(v)$ equals the number of min-cost $v$-trees in $G[X]$. Given that this is true, one can show that $w_X(\mathcal{B}_{Y,\{s\}}(G_X))$ equals the number of min-cost branchings in $G$ that connect any node in $v \in Y$ to the sink $s$ and that $w_X(\mathcal{B}(G_X))$ equals the number of all min-cost branchings in $G$. Hence, we can utilize the Markov chain tree theorem on the graph $G_X$ to compute

---

**Algorithm 2** Computation of Mixed Borda Branching

---
1: compute $(\mathcal{F}, E_y, y)$, set $\mathcal{F}' = \mathcal{F} \cup \{N \cup S\}$ and label its elements "unprocessed" ▷ Algorithm 1
2: $t_{\{v\}}(v) \leftarrow 1$ for all $v \in N \cup S$, label singletons as "processed"
3: **do** pick unprocessed $X \in \mathcal{F}'$ for which all children are processed, label $X$ as "processed"
4:      set $w_X(Y, Y') \leftarrow \sum_{(u,v) \in E_y \cap (Y \times Y')} t_Y(u)$, for all children $Y$ and $Y'$ of $X$
5:      **if** $X \neq N \cup S$ **then**
6:          **for** all children $Y$ of $X$ **do**
7:              $t_X(v) \leftarrow w_X(\mathcal{T}_Y(G_X)) \cdot t_Y(v)$ for all $v \in Y$          ▷ Lemma 1
8:      **else** compute absorbing probability matrix $Q$ for the Markov chain      ▷ Lemma 3
             corresponding to $(G_X, w_X)$
9: **while** $X \neq N \cup S$
10: **return** for all $v \in N$ and $s \in S$: $A_{v,s} \leftarrow Q_{Y_v, \{s\}}$, where $Y_v$ child of $N \cup S$ with $v \in Y_v$

---

the outcome of Mixed Borda Branching.

As for the running time, the main observation is that the computation of the number of branchings in a weighted digraph can be done in time logarithmic in the highest weight of an edge (and polynomial in the number of edges). Since all of our weights are bounded by the maximum number of branchings in the original graph (which is bounded by $|N|^{|N|}$), we can show the polynomial running time. $\qquad\square$

## 6 Equivalence of Mixed Borda Branching and Random Walk Rule

With the help of the Markov chain tree theorem, as stated in Section 5, we can show the equivalence of the Random Walk Rule and Mixed Borda Branching.

**Theorem 5.** *Let $(G, c)$ be a delegation graph and $A$ and $\hat{A}$ be the assignments returned by* Mixed Borda Branching *and the* Random Walk Rule*, respectively. Then, $A = \hat{A}$.*

*Proof.* Let $v \in N$ and $s \in S$, then,

$$
\hat{A}_{v,s} = \lim_{\varepsilon \to 0} \Big( \frac{1}{\tau} \sum_{\tau=1}^{\infty} (P^{(\varepsilon)})^\tau \Big)_{v,s} = \lim_{\varepsilon \to 0} \frac{\sum_{B \in \mathcal{B}_{v,s}} \prod_{e \in B} P_e^{(\varepsilon)}}{\sum_{B \in \mathcal{B}} \prod_{e \in B} P_e^{(\varepsilon)}} = \lim_{\varepsilon \to 0} \frac{\sum_{B \in \mathcal{B}_{v,s}} \prod_{(u,v) \in B} \frac{\varepsilon^{c(u,v)}}{\bar{\varepsilon}_u}}{\sum_{B \in \mathcal{B}} \prod_{(u,v) \in B} \frac{\varepsilon^{c(u,v)}}{\bar{\varepsilon}_u}}
$$

$$
= \lim_{\varepsilon \to 0} \frac{\prod_{u \in N} (\bar{\varepsilon}_u)^{-1} \sum_{B \in \mathcal{B}_{v,s}} \varepsilon^{c(B)}}{\prod_{u \in N} (\bar{\varepsilon}_u)^{-1} \sum_{B \in \mathcal{B}} \varepsilon^{c(B)}} = \frac{\sum_{B \in \mathcal{B}_{v,s}^*} 1}{\sum_{B \in \mathcal{B}^*} 1} = A_{v,s}.
$$

We first use the definition of the Random Walk Rule, and then apply the Markov chain tree theorem (Lemma 3) for fixed $\varepsilon \in (0, 1]$ to obtain the second equality. For the third equality, we use the definition of $P^{(\varepsilon)}$, and then factor out the normalization factor $\bar{\varepsilon}_u$ for every $u \in N$. For doing so, it is important to note that for every $v \in N$ and $s \in S$, every branching in $\mathcal{B}_{v,s}$ and every branching in $\mathcal{B}$ contains exactly one outgoing edge per node in $N$. We also remind the reader that $c(B) = \sum_{e \in B} c(e)$. Finally, we resolve the limit in the fifth equality by noting that the dominant parts of the polynomials are those corresponding to min-cost (Borda) branchings. $\qquad\square$

Given Theorem 5, we can interpret Algorithm 2 as an algorithm computing limit absorbing probabilities of a class of parametric Markov chains. We explain this reinterpretation in Appendix B.

## 7 Axiomatic Analysis

In this section, we generalize and formalize the axioms mentioned in Section 1 and show that the Random Walk Rule (and hence Mixed Borda Branching) satisfies all of them. In particular, our version of confluence (copy-robustness, respectively) reduces to (is stronger than, respectively) the corresponding axiom for the non-fractional case by Brill et al. [2022] (see Appendix C.1). We first define *anonymity*, which prescribes that a delegation rule should not make decisions based on the identity of the voters. Given a digraph with a cost function $(G, c)$ and a bijection $\sigma : V(G) \to V(G)$, we define the graph $\sigma((G, c))$ as $(G', c')$, where $V(G') = V(G)$, $E(G') = \{(\sigma(u), \sigma(v)) \mid (u, v) \in E(G)\}$ and $c'(\sigma(u), \sigma(v)) = c(u, v)$ for each edge $(u, v) \in E(G)$.

**Anonymity:** For any delegation graph $(G, c)$, any bijection $\sigma : V(G) \rightarrow V(G)$, and any $v \in N, s \in S$, it holds that $A_{v,s} = A'_{\sigma(v),\sigma(s)}$, where $A$ and $A'$ are the outputs of the delegation rule for $(G, c)$ and $\sigma((G, c))$, respectively.

**Theorem 6 (★).** *The* RANDOM WALK RULE *satisfies anonymity.*

We now define *copy-robustness*, which intuitively demands that if a delegating voter $v \in N$ decides to cast their vote instead, the total voting weight of $v$ and all its representatives should not change. This lowers the threat of manipulation by a voter deciding whether to cast or delegate their vote depending on which gives them and their representatives more total voting weight. This axiom was introduced (under a different name) by Behrens and Swierczek [2015] and defined for non-fractional delegation rules in [Brill et al., 2022]. We strengthen[10] and generalize the version of Brill et al. [2022].

**Copy-robustness:** For every delegation graph $(G, c)$ and delegating voter $v \in N$, the following holds: Let $(\hat{G}, c)$ be the graph derived from $(G, c)$ by removing all outgoing edges of $v$, let $A$ and $\hat{A}$ be the output of the delegation rule for $(G, c)$ and $(\hat{G}, c)$, respectively and let $S_v = \{s \in S \mid A_{v,s} > 0\}$ be the set of representatives of $v$ in $(G, c)$. Then $\sum_{s \in S_v} \pi_s(A) = \pi_v(\hat{A}) + \sum_{s \in S_v} \pi_s(\hat{A})$.

**Theorem 7 (★).** *The* RANDOM WALK RULE *satisfies copy-robustness.*

*Proof sketch.* We show a statement that is slightly stronger than the condition for copy-robustness. Namely, the voting weight of every casting voter from $S \setminus S_v$ remains equal when $v$ changes from being a delegating voter to becoming a casting voter. Let $(G_X, w_X)$, with $X = N \cup S$, be the contracted graph constructed in Algorithm 2 for $(G, c)$. Analogously, let $(\hat{G}_X, \hat{w}_X)$ be the contracted graph constructed for $(\hat{G}, c)$. In order to argue about their relation, we first show: If $y$ and $\hat{y}$ are the functions returned by Algorithm 1 for $G$ and $\hat{G}$, respectively, then $\hat{y}(Y) = y(Y)$ for every set $Y \subseteq 2^{N \setminus \{v\}}$ not containing $v$, $\hat{y}(\{v\}) = 1$, and $\hat{y}(Y) = 0$ for all other sets containing $v$. Now, let $Y_v$ be the node in $G_X$ containing $v$ and let $\mathcal{U} \subseteq V(G_X)$ be the subset of nodes in $G_X$ that are not reachable from $Y_v$. We argue that for any $s \in S \setminus S_v$, the Markov chain induced by $(G_X, w_X)$ can only reach $\{s\}$ from a starting node in $\mathcal{U}$. However, for all nodes in $\mathcal{U}$, all of their walks to any $\{s\}, s \in S \setminus S_v$ are still existent in the graph $(\hat{G}_X, \hat{w}_X)$, and still have the same weight. Hence, the voting weight of any $s \in S \setminus S_v$ remains unchanged when moving from $G$ to $\hat{G}$. $\qquad\square$

To capture the requirement that the voting weight of different voters is assigned to casting voters in a "consistent" way, Brill et al. [2022] define confluence as follows: A delegation rule selects, for every voter $u \in N$, one walk in the delegation graph starting in $u$ and ending in some sink $s \in S$, and assigns voter $u$ to casting voter $s$. A delegation rule satisfies confluence, if, as soon as the walk of $u$ meets some other voter $v$, the remaining subwalk of $u$ equals the walk of $v$.[11] Below, we provide a natural generalization of the property by allowing a delegation rule to specify a probability distribution over walks. Then, conditioned on the fact that the realized walk of some voter $u$ meets voter $v$, the probability that $u$ reaches some sink $s \in S$ should equal the probability that $v$ reaches $s$.

**Confluence:** For every delegation graph $(G, c)$, there exists a probability distribution $f_v$ for all $v \in N$ over the set of walks in $G$ that start in $v$ and end in some sink, which is consistent with the assignment $A$ of the delegation rule (i.e., $\mathbb{P}_{W \sim f_v}[s \in W] = A_{v,s}$ for all $v \in N, s \in S$), and,

$$\mathbb{P}_{W \sim f_u}[s \in W \mid v \in W] = \mathbb{P}_{W \sim f_v}[s \in W] \quad \text{for all } u, v \in N, s \in S.$$

Note that the requirement that $A_{v,s} = \mathbb{P}_{W \sim f_v}[s \in W]$ implies that for any $v \in V$ and $s \in S$ we can have $A_{v,s} > 0$ only if there is a path from $v$ to $s$ in $G$.

**Theorem 8 (★).** *The* RANDOM WALK RULE *satisfies confluence.*

*Proof sketch.* One can verify that every delegation rule that can be formalized via a Markov chain on the delegation graph $(G, c)$ satisfies confluence. In Section 5, we showed that the RANDOM WALK RULE can be computed by solving a Markov chain $(G_X, P)$ on the contracted graph $G_X$ (for $X = N \cup S$). We utilize $(G_X, P)$ to define a probability distribution over walks in $G$ that satisfies confluence: Every walk in $G$ can be mapped to a walk in $G_X$ (by ignoring edges inside children of $X$), but there may exist many walks in $G$ that map to the same walk in $G_X$. We pick, for every walk

---

[10]Brill et al. [2022] restrict the condition to voters that have a direct connection to their representative.

[11]Brill et al. [2022] assume paths instead of walks. The two definitions are equivalent (see Appendix C.1).

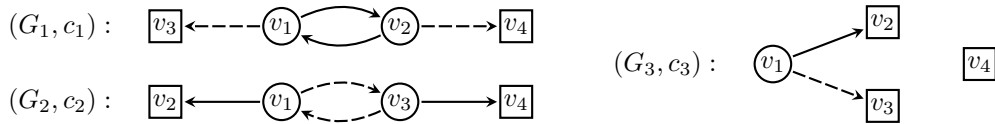

Figure 3: Situation in the proof of Theorem 9. Solid edges correspond to first-choice delegations, dashed edges to second-choice delegations.

$\hat{W}$ in $G_X$, one representative walk $W$ in $G$ and give it the same probability as $\hat{W}$ in $(G_X, P)$. All other walks in $G$ get zero probability. When picking the representative walks, we can ensure that for every two nodes $u, v \in N \cup S$, the probability that the walk of $u$ reaches $v$ equals the probability that $Y_u$ reaches $Y_v$ in $(G_X, P)$, where $Y_u$ and $Y_v$ are the nodes in $V(G_X)$ containing $u$ and $v$, respectively. For the constructed probability distribution, we then show that the confluence condition is met. $\qquad\square$

With the formal definition of confluence, anonymity, and copy-robustness we can now show that these properties altogether are impossible to achieve in the non-fractional case. Recall, that a non-fractional delegation rule is defined as a delegation rule, that returns assignments $A \in \{0, 1\}^{N \times S}$.

**Theorem 9.** *No non-fractional delegation rule satisfies confluence, anonymity, and copy-robustness.*

*Proof.* Consider the graph $(G_1, c_1)$ in Figure 3. There are four non-fractional assignments in $(G_1, c_1)$: Both $v_1$ and $v_2$ can either be assigned $v_3$ or $v_4$. Suppose a rule chooses assignment $A$ with $A_{v_1,v_4} = A_{v_2,v_3} = 1$. This rule cannot satisfy confluence, as any walk from $v_2$ to $v_3$ includes $v_1$ and confluence requires $1 = A_{v_2,v_3} = \mathbb{P}_{W \sim f_{v_2}}[v_3 \in W] = \mathbb{P}_{W \sim f_{v_2}}[v_3 \in W \mid v_1 \in W] = \mathbb{P}_{W \sim f_{v_1}}[v_3 \in W] = A_{v_1,v_3} = 0$. Now, suppose a delegation rule chooses assignment $A$ with $A_{v_1,v_3} = A_{v_2,v_3} = 1$. We define the bijection $\sigma$ mapping $v_1$ to $v_2$, $v_2$ to $v_1$, $v_3$ to $v_4$ and $v_4$ to $v_3$. Then, $\sigma((G_1, c)) = (G_1, c)$ and thus $A'_{v_1,v_3} = A'_{v_2,v_3} = 1$ in the assignment $A'$ that the rule chooses for $\sigma((G_1, c))$. This contradicts anonymity, since $1 = A_{v_1,v_3} \neq A'_{\sigma(v_1),\sigma(v_3)} = A'_{v_2,v_4} = 0$. We can make the same argument in the case of $A_{v_1,v_4} = A_{v_2,v_4} = 1$. For any rule satisfying anonymity and confluence the chosen assignment $A$ must therefore have $A_{v_1,v_3} = A_{v_2,v_4} = 1$.
The above arguments are independent of the cost function $c$, so long as we have $c(v_1, v_2) = c(v_2, v_1)$ and $c(v_1, v_3) = c(v_2, v_4)$, needed for the equality of $\sigma((G_1, c))$ and $(G_1, c)$. Thus, any rule satisfying anonymity and confluence must choose the assignment $A$ with $A_{v_1,v_2} = A_{v_3,v_4} = 1$ for $(G_2, c_2)$.
We modify $(G_1, c_1)$ by making $v_2$ a casting voter (as in the definition of copy-robustness) and retrieve $(G_3, c_3)$. Copy robustness requires that the assignment from $v_1$ to $v_4$ in $G_1$ (which is zero) must be the same as the sum of assignments from $v_1$ to $v_4$ and $v_2$. Thus, we have $A_{v_1,v_3} = 1$ for the assignment $A$, that any confluent, anonymous, and copy-robust rule chooses for $(G_3, c_3)$. However, we can also construct $(G_3, c_3)$ from $(G_2, c_2)$ by making $v_3$ a casting voter. Then, analogously, copy-robustness requires $A_{v_1,v_2} = 1$ for the assignment of $(G_3, c_3)$, leading to a contradiction. $\qquad\square$

Since the RANDOM WALK RULE (and thus MIXED BORDA BRANCHING) satisfies generalizations of the three axioms, the above impossibility is due to its restriction to non-fractional rules.

## 8 Concluding Remarks

We generalized the setting of liquid democracy with ranked delegations to allow for fractional delegation rules. Beyond that, we presented a delegation rule that can be computed in polynomial time and satisfies a number of desirable properties. A natural follow-up question is to understand the entire space of delegation rules satisfying these properties.
Fractional delegations have been recently implemented (see electric.vote) and studied by Degrave [2014] and Bersetche [2022]. In contrast to our setting, these approaches let agents declare a desired *distribution* over their delegates (instead of rankings). We remark that one could easily combine the two approaches by letting agents declare their desired split within each equivalence class of their ranking. Our algorithm can be extended for this setting (see Appendix B).
There exists a line of research which aims to understand liquid democracy from an epistemic viewpoint [Kahng et al., 2021, Caragiannis and Micha, 2019, Halpern et al., 2023]. Here, many of the negative results stem from the fact that voting weight is concentrated on few casting voters. Since, intuitively, ranked delegations can help to distribute the voting weight more evenly, it would be interesting to study these through the epistemic lens.

## Acknowledgements

This work was supported by the *Deutsche Forschungsgemeinschaft* (under grant BR 4744/2-1), the *Centro de Modelamiento Matemático (CMM)* (under grant FB210005, BASAL funds for center of excellence from ANID-Chile), *ANID-Chile* (grant ACT210005), and the *Dutch Research Council (NWO)* (project number 639.023.811, VICI "Collective Information"). Moreover, this work was supported by the National Science Foundation under Grant No. DMS-1928930 and by the Alfred P. Sloan Foundation under grant G-2021-16778, while Ulrike Schmidt-Kraepelin was in residence at the Simons Laufer Mathematical Sciences Institute (formerly MSRI) in Berkeley, California, during the Fall 2023 semester.

We would like to thank Markus Brill for suggesting the setting to us as well as insightful discussions. Moreover, we thank Jannik Matuschke for helpful discussions on min-cost branchings. Also, we thank our colleagues from Universidad de Chile, Martin Lackner, and Théo Delemazure for their valuable feedback.

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

# Appendix

## A    Missing Proofs of Section 5

**Lemma 2 (★).** *Let $(G, c)$ be a delegation graph and let $(\mathcal{F}, E_y, y)$ be the output of Algorithm 1. Then:*

   *(i) For every $(G, c)$, the output of the algorithm is unique, i.e., it does not depend on the choice of the strongly connected component in line 3.*

   *(ii) $\mathcal{F}$ is laminar, i.e., for any $X, Y \in \mathcal{F}$ it holds that either $X \subseteq Y$, $Y \subseteq X$, or $X \cap Y = \emptyset$.*

   *(iii) Branching $B$ in $(G, c)$ is min-cost iff (a) $B \subseteq E_y$, and (b) $|B \cap \delta^+(X)| = 1$ for all $X \in \mathcal{F}, X \subseteq N$.*

   *(iv) For every $X \in \mathcal{F}$, an in-tree $T$ in $G[X] = (X, E[X])$, where $E[X] = \{(u, v) \in E \mid u, v \in X\}$, is min-cost iff (a) $T \subseteq E_y$, and (b) $|T \cap \delta^+(Y)| = 1$ for all $Y \in \mathcal{F}$ such that $Y \subset X$.*

*Proof.*    Let $(G, c)$ be a delegation graph and let $(\mathcal{F}, E_y, y)$ be the output of Algorithm 1.

We start by proving statement (ii). The sets in $\mathcal{F}$ correspond exactly to those sets with positive $y$-value. Assume for contradiction that there exist two sets $X, Y \in \mathcal{F}$ with $X \cap Y \neq \emptyset$ and none of the subsets is a subset of the other. Assume without loss of generality that $X$ was selected before $Y$ by the algorithm and let $y_1$ and $y_2$ be the status of the function $y$ in each of the two situation. Then, by construction of the algorithm it holds that $G_1 = (N \cup S, E_{y_1})$ is a subgraph of $G_2 = (N \cup S, E_{y_2})$. This is because once an edge is added to the set of tight edges (denoted by $E_y$) it remains in this set. Since $Y$ is a strongly connected component in $G_2$ without outgoing edge, it holds that for every $z \in X \setminus Y$ and $z' \in X \cap Y$, the node $z'$ does not reach $z$ in $G_2$. However, this is a contradiction to the fact that $X$ is a strongly connected component in the graph $G_1$, which concludes the proof of statement (ii).

We now turn to prove statement (i) and already assume that $\mathcal{F}$ is laminar. We fix an order of the selected strongly connected components in line 3 of the algorithm. Then, suppose that for some other choices in line 3, the algorithm returns some other output $(\hat{\mathcal{F}}, E_{\hat{y}}, \hat{y})$. Note that $\hat{\mathcal{F}} \neq \mathcal{F}$ or $E_{\hat{y}} \neq E_y$ implies that $\hat{y} \neq y$. Thus, it suffices to assume for contradiction that $\hat{y} \neq y$. Then there must be a smallest set $X$, that has $y(X) \neq \hat{y}(X)$ (without loss of generality we assume $y(X) > \hat{y}(X)$). Let $\mathcal{X} = 2^X \setminus \{X\}$ be the set of all strict subsets of $X$. Since we defined $X$ to be of minimal cardinality, we have $y[\mathcal{X}] = \hat{y}[\mathcal{X}]$, where $y[\mathcal{X}]$, and $\hat{y}[\mathcal{X}]$ denote the restriction of $y$ and $\hat{y}$ to $\mathcal{X}$, respectively. Because $y(X) > 0$, all children of $X$ are strongly connected by tight edges with respect to $y[\mathcal{X}]$ and have no tight edges pointing outside of $X$. Now, consider the iteration of the alternative run of the Algorithm 2, in which the algorithm added the last set in $\mathcal{X} \cup \{X\}$. Since $\hat{y}(X) < y(X)$, for every further iteration of the algorithm, a chosen set $X' \neq X$ cannot contain any node in $X$ (because otherwise $X'$ cannot form a strongly connected component without outgoing edge). However, since the nodes in $X$ cannot reach a sink via tight edges, this is a contradiction to the termination of the algorithm.

We now prove statement (iii). The plan of attack is the following: First we define a linear program that captures the min-cost branchings in a delegation graph. Second, we dualize the linear program and show that $y$ (more precisely a minor variant of $y$) is an optimal solution to the dual LP, and third, utilize complementary slackness to prove the claim. For a given delegation graph $(G, c)$ with $V(G) = N \cup S$ we define the following linear program, also denoted by (LP):

$$\min \sum_{e \in E} c(e) x_e$$
$$\sum_{e \in \delta^+(X)} x_e \geq 1 \qquad\qquad \forall\, X \subseteq N$$
$$x_e \geq 0 \qquad\qquad \forall e \in E$$

We claim that every branching $B$ in $G$ induces a feasible solution to (LP). More precisely, given a branching $B$, let

$$x_e = \begin{cases} 1 & \text{if } e \in B \\ 0 & \text{if } e \notin B. \end{cases}$$

The last constraint is trivially satisfied. Now, assume for contradiction that there exists $X \subseteq N$ such that the corresponding constraint in (LP) is violated. In this case the nodes in $X$ have no path towards some sink node in $B$, a contradiction to the fact that $B$ is a (maximum cardinality) branching. In particular, this implies that the objective value of (LP) is at most the minimum cost of any branching in $G$ (in fact the two values are equal, but we do not need to prove this at this point). We continue by deriving the dual of (LP), to which we refer to as (DLP):

$$\max \sum_{X \subseteq N} y_X$$
$$\sum_{X \subseteq N | e \in \delta^+(X)} y_X \le c(e) \qquad \forall\, e \in E$$
$$y_X \ge 0 \qquad \forall X \subseteq N$$

Now, let $y$ be the function returned by Algorithm 1. We define $\hat{y}$, which is intuitively $y$ restricted to all subsets on $N$, more precisely, $\hat{y}(X) = y(X)$ for all $X \subseteq N$. We claim that $\hat{y}$ is a feasible solution to (DLP). This can be easily shown by induction. More precisely, we fix any $e \in E$ and show that the corresponding constraint in (DLP) is satisfied throughout the execution of the algorithm. At the beginning of the algorithm $y$ (and hence $\hat{y}$) is clearly feasible for (DLP). Now, consider any step in the algorithm and let $X$ be the selected strongly connected component. If $e \in \delta^+(X)$, then we know that the constraint corresponding to $e$ is not tight (since $X$ has no tight edge in its outgoing cut). Moreover, $y$ is only increased up to the point that some edge in $\delta^+(X)$ becomes tight (and not higher than that). Hence, after this round, the constraint for $e$ is still satisfied. If, on the other hand, $e \notin \delta^+(X)$, then the left-hand-side of $e$'s constraint remains equal when $y(X)$ is increased. Hence, the constraint of $e$ is still satisfied.

Next, we claim that there exists a branching $B$ in $G$, such that for the resulting primal solution $x$, it holds that $\sum_{e \in E} c(e)x_e = \sum_{X \subseteq N} \hat{y}(X)$. The branching $B$ will be constructed in a top-down fashion by moving along the laminar hierarchy of $\mathcal{F}$. To this end let $G_X$ be the contracted graph as defined in Algorithm 2. We start by setting $X = N \cup S$. Since every node in $N$ can reach some sink via tight edges, we also know that every node in $G_X$ can reach some sink. Hence, a branching in $G_X$ has exactly one edge per node in $V_X$ that is not a sink. Let's pick such a branching $B_X$. We know that for every edge in $B_X = (Y, Z)$ there exists some edge in the original graph $G$ that is also tight, i.e., $u \in Y$ and $v \in Z$ such that $(u, v) \in E_y$. For every edge in $B_X$ pick an arbitrary such edge and add it to $B$. Now, pick an arbitrary node $Y \in V(G_X)$. By construction, we know that exactly one edge from $B$ is included in $\delta^+(Y)$, call this edge $(u, v)$. Then, within the graph $G_Y$, there exists exactly one node $Z \in V(G_Y)$, that contains $u$. We are going to search for a $Z$-tree within $G_Y$. We know that such a tree exists since $G_Y$ is strongly connected by construction. We follow the pattern from before, i.e., finding a $Z$-tree, mapping the edges back to the original graph (arbitrarily), and then continuing recursively. For proving our claim, it remains to show that $\sum_{e \in E} c(e)x_e = \sum_{X \subseteq N} \hat{y}(X)$. The crucial observation is that, by construction, every set in $\hat{\mathcal{F}} = \mathcal{F} \setminus \{\{s\} \mid s \in S\}$ is left by exactly one edge in $B$. Hence, we can partition the set $\hat{\mathcal{F}}$ into sets $\bigcup_{e \in B} \hat{F}_e$, where $\hat{\mathcal{F}}_e = \{X \in \hat{\mathcal{F}} \mid e \in \delta^+(X)\}$. Moreover, observe that every edge in $B$ is tight. As a result we get that

$$\sum_{e \in B} c(e)x_e = \sum_{e \in B} \sum_{X \in \hat{\mathcal{F}}_e} \hat{y}(X) = \sum_{X \subseteq N} \hat{y}(X),$$

proving the claim.

As a result, note that we found a primal solution $B$ (precisely, the $x$ induced by $B$), and a dual solution $\hat{y}$ having the same objective value. By weak duality, we can conclude that both solutions are in particular optimal. It only remains to apply complementary slackness to conclude the claim. To this end, let $B$ be a min-cost branching and $x$ be the induced primal solution. By the argument above we know that $x$ is optimal. Now, for any $X \subseteq N$ for which $\hat{y}(X) > 0$ (hence $X \in \mathcal{F}$), complementary slackness prescribes that the corresponding primal constraint is tight, i.e., $\sum_{e \in \delta^+(X)} x_e = 1$. Hence, the branching corresponding to $x$ leaves the set $X$ exactly once, and part (b) of statement (iii) is

satisfied. For statement (a) we apply complementary slackness in the other direction. That is, when $x_e > 0$, this implies that the corresponding dual constraint is tight, implying that $e$ has to be tight with respect to $\hat{y}$ and therefore also with respect to $y$ (recall that $y$ and $\hat{y}$ only differ with respect to the sink nodes).

We now turn to proving statement (iv). This is done almost analogously to statement (iii). Fix $X \in \mathcal{F}$. In the following we argue about the min-cost in-trees in $G[X]$ and how to characterize these via a linear program. To this end, we add a dummy sink node $r$ to the graph $G[X]$ and call the resulting graph $\hat{G}$. More precisely, $\hat{G} = (X \cup \{r\}, E[X] \cup \{(u, r) \mid u \in X\})$. The cost of any edge $(u, r), u \in X$ is set to $c^* := \max_{e \in E(G)} c(e) + 1$, where it is only important that this value is larger than any other cost in the graph. We define the following LP:

$$\min \sum_{e \in E(\hat{G})} c(e) x_e$$

$$\sum_{e \in \delta^+_{\hat{G}}(Z)} x_e \geq 1 \qquad\qquad \forall\, Z \subseteq X$$

$$x_e \geq 0 \qquad\qquad \forall e \in E(\hat{G})$$

For every min-cost in-tree $T$ in $G[X]$ we obtain a feasible solution to (LP). To this end, let $u \in X$ be the sink node of $T$ and define $\hat{T} = T \cup \{(u, r)\}$. Then, translate $\hat{T}$ to its incidence vector $x$. Given this observation, we again derive the dual of (LP), to which we refer to as (DLP):

$$\max \sum_{Z \subseteq X} y_Z$$

$$\sum_{Z \subseteq X \mid e \in \delta^+_{\hat{G}}(Z)} y_Z \leq c(e) \qquad\qquad \forall\, e \in E(\hat{G})$$

$$y_Z \geq 0 \qquad\qquad \forall Z \subseteq X$$

Now, let $y$ be the output of Algorithm 1 for the original graph $G$. We derive $\hat{y} : 2^X \to \mathbb{R}$ as follows:

$$\hat{y}(Z) = \begin{cases} y(Z) & \text{if } Z \subset X \\ c^* - \max_{u \in X} \sum_{Z \subset X \mid e \in \delta^+_{\hat{G}}(Z)} y(Z) & \text{if } Z = X \end{cases}$$

First, analogously to (iii), it can be verified that $\hat{y}$ is a feasible solution to (DLP). Moreover, again analogously to (iii), there exists some min-cost $r$-tree in $\hat{G}$ and a corresponding primal solution $x$, such that $\sum_{e \in E(\hat{G})} c(e) x_e = \sum_{Z \subseteq X} \hat{y}_X$. (This tree is derived by first chosing a tight edge towards the dummy root node $r$ and then again recurse over the laminar family $\mathcal{F}$ restricted to $X$.) This implies by weak duality that $\hat{y}$ is an optimal solution to (DLP) and any min-cost $r$-tree in $\hat{G}$ is an optimal solution to (LP). As a result, we can again apply complementary slackness in both directions: Let $T$ be a min-cost in-tree in $G[X]$ with sink node $u \in X$. Then let $\hat{T} = T \cup \{(u, r)\}$ be the corresponding min-cost $r$-tree in $\hat{G}$ and $x$ be the corresponding incidence vector. Then, complementary slackness implies that for any $e \in E[X]$ for which $x_e > 0$ (and hence $e \in T$), it holds that the corresponding constraint in (DLP) is tight with respect to $\hat{y}$ (and also $y$). This implies that $e \in E_y$. On the other hand, for any $Z \subset X$, if $\hat{y}_Z > 0$, and hence $X \in \mathcal{F}$, complementary slackness prescribes that the corresponding primal constraint is tight, and hence $|T \cap \delta^+_{G[X]}(Z)| = 1$, concluding the proof. $\square$

For the proof of the next theorem, we first explain how to compute the absorbing probabilities of an absorbing Markov chain $(G, P)$ and show a related lemma that we need in Appendix C. W.l.o.g. we assume that the states $V(G)$ are ordered such that the non-absorbing states $N$ come first and the absorbing states $S$ last. We can then write the transition matrix as

$$P = \begin{bmatrix} D & C \\ 0 & I_{|S|} \end{bmatrix} \quad,$$

where $D$ is the $|N| \times |N|$ transition matrix from non-absorbing states to non-absorbing states and $C$ is the $|N| \times |S|$ transition matrix from non-absorbing states to absorbing states. $I_{|S|}$ denotes the $|S| \times |S|$ identity matrix. The absorbing probability of an absorbing state $s \in S$, when starting a random walk in a state $v \in N$ is then given as the entry in the row corresponding to $v$ and the column corresponding to $s$ in the $|N| \times |S|$ matrix $(I_{|N|} - D)^{-1} C$ [Grinstead and Snell, 1997].

**Lemma A.1.** *Adding a self-loop to a non-absorbing state $v$ with probability $p$ and scaling all other transition probabilities from that state by $1 - p$ does not change the absorbing probabilities of an absorbing Markov-chain $(G, P)$.*

*Proof.* Let $(D, C)$ and $(D', C')$ be the transition matrices of the absorbing Markov chain before and after adding the self-loop. Let $\mathbf{d}_v, \mathbf{d}'_v, \mathbf{c}_v, \mathbf{c}'_v$ be the rows of $D, D', C, C'$, corresponding to state $v$ respectively. Then

$$\mathbf{d}'_v = (1 - p)\mathbf{d}_v + p\mathbf{e}_v^{\mathsf{T}} \quad ,$$
$$\mathbf{c}'_v = (1 - p)\mathbf{c}_v$$

and $\mathbf{d}'_u = \mathbf{d}_u$ and $\mathbf{c}'_u = \mathbf{c}_u$ for all $u \neq v$.

We want to show that $(I_{|N|} - D)^{-1}C = (I_{|N|} - D')^{-1}C'$. Let $Z = (I_{|N|} - D)^{-1}C$. Then $Z = (I_{|N|} - D')^{-1}C'$ if and only if $Z = D'Z + C'$. Notice, that only the row corresponding to $v$ in $D'$ and $C'$ differ from $D$ and $C$ and therefore for all $u \neq v$

$$\mathbf{z}_u = \mathbf{d}_u Z + \mathbf{c}_u = \mathbf{d}'_u Z + \mathbf{c}'_u \quad ,$$

where $\mathbf{z}_u$ is the row of $Z$ corresponding to $u$. The only thing left to show is $\mathbf{z}_v = \mathbf{d}'_v Z + \mathbf{c}'_v$. We have

$$
\begin{aligned}
\mathbf{d}'_v Z + \mathbf{c}'_v &= ((1 - p)\mathbf{d}_v + p\mathbf{e}_v^{\mathsf{T}})Z + (1 - p)\mathbf{c}_v \\
&= (1 - p)\mathbf{d}_v Z + p\mathbf{e}_v^{\mathsf{T}} Z + (1 - p)\mathbf{c}_v \\
&= (1 - p)(\mathbf{d}_v Z + \mathbf{c}_v) + p\mathbf{e}_v^{\mathsf{T}} Z \\
&= (1 - p)\mathbf{z}_v + p\mathbf{z}_v \qquad\qquad \text{(since } DZ + C = Z) \\
&= \mathbf{z}_v \quad ,
\end{aligned}
$$

which concludes the proof. $\qquad\square$

**Theorem 4 (★).** *Algorithm 2 returns* MIXED BORDA BRANCHING *and runs in* $poly(n)$.

*Proof.* We start by showing by induction that the given interpretation of the weight function on the nodes is correct, i.e., for any $v \in N$, $t_X(v)$ corresponds to the number of min-cost $v$-trees in the graph $G[X]$. The claim is clearly true for any singleton, since $t_{\{v\}}(v) = 1$ and the number of $v$-trees in $(\{v\}, \emptyset)$ is one, i.e., the empty set is the only $v$-trees. Now, we fix some $X \in \mathcal{F}'$ and assume that the claim is true for all children of $X$. In the following, we fix $v \in X$ and argue that the induction hypothesis implies that the claim holds for $t_X(v)$ as well.

For any node $u \in X$, let $Y_u \in \mathcal{F}$ be the child of $X$ containing node $u$. Moreover, let $\mathcal{T}_v^*(G[X])$ (or short $\mathcal{T}_v^*$) be the set of min-cost $v$-trees in $G[X]$, and $\mathcal{T}_{Y_v}(G_X)$ (or short $\mathcal{T}_{Y_v}$) be the set of $Y_v$-trees in $G_X$. Lastly, for any $u \in X$, let $\mathcal{T}_u^*(G[Y_u])$ be the set of min-cost $u$-trees in $G[Y_u]$. In the following, we argue that there exists a many-to-one mapping from $\mathcal{T}_v^*$ to $\mathcal{T}_{Y_v}$. Note that, by statement (iv) in Lemma 2, every min-cost in-tree $T$ in $G[X]$ (hence, $T \in \mathcal{T}_v^*$) leaves every child of $X$ exactly once via a tight edge. Therefore, there exists a natural mapping to an element of $\mathcal{T}_{Y_v}$ by mapping every edge in $T$ that connects two children of $X$ to their corresponding edge in $G_X$. More precisely, $\hat{T} = \{(Y, Y') \in E_X \mid T \cap \delta^+(Y) \cap \delta^-(Y') \neq \emptyset\}$ is an $Y_v$-tree in $G_X$ and hence an element of $\mathcal{T}_{Y_v}$.

Next, we want to understand how many elements of $\mathcal{T}_v^*$ map to the same element in $\mathcal{T}_{Y_v}$. Fix any $\hat{T} \in \mathcal{T}_{Y_v}$. We can construct elements of $\mathcal{T}_v^*$ by combining (an extended version of) $\hat{T}$ with min-cost in-trees within the children of $X$, i.e., with elements of the sets $\mathcal{T}_u^*(G[Y_u])$, $u \in X$. More precisely, for any edge $(Y, Y') \in \hat{T}$, we can independently chose any of the edges in $(u, u') \in E_y \cap (Y \times Y')$ and combine it with any min-cost $u$-tree in the graph $G[Y]$. This leads to

$$
\left( \prod_{(Y,Y') \in \hat{T}} \sum_{(u,u') \in E_y \cap (Y \times Y')} t_Y(u) \right) t_{Y_v}(v) = \left( \prod_{(Y,Y') \in \hat{T}} w_X(Y, Y') \right) \cdot t_{Y_v}(v)
$$

many different elements from $\mathcal{T}_v^*$ that map to $\hat{T} \in \mathcal{T}_{Y_v}$. Hence,

$$
|\mathcal{T}_v^*| = \sum_{\hat{T} \in \mathcal{T}_{Y_v}} \prod_{(Y,Y') \in \hat{T}} w_Y(Y, Y') \cdot t_{Y_v}(v) = w_X(\mathcal{T}_{Y_v}) \cdot t_{Y_v}(v) = t_X(v),
$$

where the last inequality follows from the definition of $t_X(v)$ in the algorithm. This proves the induction step, i.e., $t_X(v)$ corresponds to the number of min-cost $v$-trees in the graph $G[X]$.

Now, let $X = N \cup S$, i.e., we are in the last iteration of the algorithm. Due to an analogous reasoning as before, there is a many-to-one mapping from the min-cost branchings in $G$ to branchings in $G_X$. More precisely, for every branching $B \in \mathcal{B}_{Y,\{s\}}(G_X)$, there exist

$$\prod_{(Y,Y')\in B} w_X(Y,Y') = w_X(B)$$

branchings in $G$ that map to $B$. Hence, by the Markov chain tree theorem (Lemma 3), we get

$$A_{v,s} = Q_{v,s} = \frac{\sum_{B \in \mathcal{B}_{Y_v,\{s\}}(G_X)} w_X(B)}{\sum_{B \in \mathcal{B}(G_X)} w_X(B)} = \frac{\sum_{B \in \mathcal{B}_{v,s}^*(G)} 1}{\sum_{B \in \mathcal{B}^*(G)} 1} \quad ,$$

where $(G'_X, P)$ is the Markov chain corresponding to $G_X$ and $Q = \lim_{\tau \to \infty} \frac{1}{\tau} \sum_{i=0}^{\tau} P^\tau$. This equals the definition of MIXED BORDA BRANCHING.

Lastly, we argue about the running time of the algorithm. For a given delegation graph $(G, c)$, let $n = V(G)$, i.e., the number of voters. Algorithm 1 can be implemented in $\mathcal{O}(n^3)$. That is because, the while loop runs for $\mathcal{O}(n)$ iterations (the laminar set family $\mathcal{F}$ can have at most $2n - 1$ elements), and finding all strongly connected components in a graph can be done in $\mathcal{O}(n^2)$ (e.g., with Kosaraju's algorithm [Hopcroft et al., 1983]). Coming back to the running time of Algorithm 2, we note that the do-while loop runs for $\mathcal{O}(n)$ iterations, again, due to the size of $\mathcal{F}'$. In line 7, the algorithm computes $\mathcal{O}(n)$ times the number of weighted spanning trees with the help of Lemma 1 (Tutte [1948]). Hence, the task is reduced to calculating the determinant of a submatrix of the laplacian matrix. Computing an integer determinant can be done in polynomial time in $n$ and $\log(M)$, if $M$ is an upper bound of all absolute values of the matrix[12]. Note, that all values in every Laplacian (the out-degrees on the diagonals and the multiplicities in the other entries) as well as the results of the computation are upper-bounded by the total number of branchings in the original graph $G$ (this follows from our argumentation about the interpretation of $t_X(v)$ in the proof of Theorem 4), hence in particular by $n^n$. Therefore, the running time of each iteration of the do-while loop is polynomial in $n$. In the final step we compute the absorbing probabilities of the (scaled down version) of the weighted graph $(G_X, w_X)$ (where $X = N \cup S$). For that, we need to compute the inverse of a $\mathcal{O}(n) \times \mathcal{O}(n)$ matrix, which can be done using the determinant and the adjugate of the matrix. Computing these comes down to computing $\mathcal{O}(n^2)$ determinants, for which we argued before that it is possible in polynomial time[13]. Summarizing, this gives us a running time of Algorithm 1 in $\mathcal{O}((n^7 \log(n) + n^4 \log(n \log(n))) * (\log^2 n + (\log(n \log(n)))^2))$. □

# B    Further Remarks on Section 6

**Alternative Interpretation of Algorithm 2**    We stated Algorithm 2 in terms of counting min-cost branchings. There exists a second natural interpretation that is closer to the definition of the RANDOM WALK RULE, in which we want to compute the limit of the absorbing probabilities of a parametric Markov chain. We give some intuition on this reinterpretation of the algorithm with the example in Figure 2, and later extend this interpretation to a larger class of parametric Markov chains.

Intuitively speaking, every set $X \in \mathcal{F}$ in the Markov chain $(G, P^{(\varepsilon)})$ corresponding to the delegation graph $G$ is a strongly connected component whose outgoing edges have an infinitely lower probability than the edges inside of $X$ as $\varepsilon$ approaches zero. We are therefore interested in the behavior of an infinite random walk in $G[X]$. While in the branching interpretation, the node weight $t_X(v)$ can be interpreted as the number of min-cost $v$-arborescences in $G[X]$, in the Markov chain interpretation we think of $t_X(v)$ as an indicator of the relative time an infinite random walk spends in $v$ (or the relative number of times $v$ is visited) in the Markov chain given by the strongly connected graph $G[X]$. Consider the example iteration depicted in Figure 2a, where we are given an unprocessed $X \in \mathcal{F}$ whose children $Y_1, Y_2$ are all processed. When contracting $Y_1$ and $Y_2$ the weights on the edges should encode how likely a transition is from one set to another, which is achieved by summing over the relative time spent in each node with a corresponding edge. We then translate the resulting

---

[12]More precisely, it can be computed in $\mathcal{O}((n^4 \log(nM) + n^3 \log^2(nM)) * (\log^2 n + (\log \log M)^2))$ [Gathen and Gerhard, 2013]

[13]We argued this only for integer matrices, but we can transform the rational matrix into an integer one by scaling it up by a factor which is bounded by $n^n$.

graph (Figure 2b) into a Markov chain and again compute the relative time spend in each node. This computation is equivalent to calculating the sum of weights of all in-trees (up to a scaling factor, see Theorem 3). Indeed, we get a ratio of one to three for the time spend in $Y_1$ and $Y_2$. To compute $t_X(v)$ we multiply the known weight $t_{Y_v}(v)$ by the newly calculated weight of $Y_v$. In the example this means that since we know, we spend three times as much time in $Y_2$ as in $Y_1$ all weights of nodes in $Y_2$ should be multiplied by three (see Figure 2c).

**Extension of Algorithm 2** In addition, we remark that our algorithm could be extended to a larger class of parametric Markov chains, namely, to all Markov chains $(G, P^{(\varepsilon)})$, where $G$ is a graph in which every node has a path to some sink node, and, for every $e \in E(G)$, $P_e^{(\varepsilon)}$ is some rational fraction in $\varepsilon$, i.e., $\frac{f_e(\varepsilon)}{g_e(\varepsilon)}$, where both $f_e$ and $g_e$ are polynomials in $\varepsilon$ with positive coefficients.[14] Now, we can construct a cost function $c$ on $G$, by setting $c(e) = x_e - z_e + 1$, where $x_e$ is the smallest exponent in $f_e(\varepsilon)$ and $z_e$ is the smallest exponent in $g_e(\varepsilon)$. Note that, if $c(e) < 1$, then the Markov chain cannot be well defined for all $\varepsilon \in (0, 1]$. Now, we run Algorithm 2 for the delegation graph $(G, c)$ with the only one difference, i.e., the weight function $w_X$ also has to incorporate the coefficients of the polynomials $f_e(\varepsilon)$ and $g_e(\varepsilon)$. More precisely, we define for every $e \in E$, the number $q_e$ as the ratio between the coefficient corresponding to the smallest exponent in $f_e$ and the coefficient corresponding to the smallest exponent in $g_e$. Then, we redefine line 4 in the algorithm to be

$$w_X(Y, Y') \leftarrow \sum_{(u,v) \in E_y \cap (Y \times Y')} t_Y(u) \cdot q_{(u,v)}.$$

One can then verify with the same techniques as in Section 5 and Section 6, that this algorithm returns the outcome of the above defined class of Markov chains.

## C  Missing Proofs and Further Results of Section 7

**Theorem 6 (★).** *The* RANDOM WALK RULE *satisfies anonymity.*

*Proof.* Given a delegation graph $(G, c)$ and a bijection $\sigma : V(G) \to V(G)$, we know that for all $v \in V(G)$ it holds that $|\delta_G^+(v)| = |\delta_{G'}^+(\sigma(v))|$ and $c(v, w) = c'(\sigma(v), \sigma(w))$ for any edge $(v, w) \in \delta^+(v)$, where $(G', c') = \sigma((G, c))$. In the corresponding Markov chains $M_\varepsilon$ and $M'_\varepsilon$ we therefore get $P_{(v,w)}^{(\varepsilon)} = P_{(\sigma(v),\sigma(w))}'^{(\varepsilon)}$ (see Equation 1). Since through the bijection between the edges of the graph, we also get a bijection between all walks in the graph $\mathcal{W}$ and for every $s \in S$ and walk in $\mathcal{W}[s, v]$ there is a corresponding walk in $\mathcal{W}[\sigma(v), \sigma(s)]$ of the same probability. Therefore we have

$$A_{v,s} = \lim_{\varepsilon \to 0} \sum_{W \in \mathcal{W}[v,s]} \prod_{e \in W} P_e^{(\varepsilon)} = \lim_{\varepsilon \to 0} \sum_{W \in \mathcal{W}[\sigma(v),\sigma(s)]} \prod_{e \in W} P_e'^{(\varepsilon)} = A'_{\sigma(v),\sigma(s)} \quad,$$

which concludes the proof. □

**Theorem 7 (★).** *The* RANDOM WALK RULE *satisfies copy-robustness.*

*Proof.* Let $(G, c)$, $v$, $(\hat{G}, c)$, $A$, $\hat{A}$ and $S_v$ be defined as in the definition of copy-robustness. Let $(\mathcal{F}, y)$ and $(\hat{\mathcal{F}}, \hat{y})$ be the set families and functions returned by Algorithm 1 for $G$ and $\hat{G}$, respectively. In this proof, we restrict our view to the subgraphs of only tight edges, denoted by $G_y = (N \cup S, E_y)$ and $\hat{G}_{\hat{y}} = (N \setminus \{v\} \cup V \cup \{v\}, E_{\hat{y}})$, respectively. Note, that this does not change the result of the RANDOM WALK RULE, since it is shown to be equal to MIXED BORDA BRANCHING, which only considers tight edges (in the contracted graph) itself.

First, we observe that the set $S_v$ is exactly the subset of $S$ reachable by $v$ in $G_y$. This is because the assignment $A$ returned by the RANDOM WALK RULE is given as the absorbing probability of a Markov chain on the graph $(G_X, w_X)$ with $X = N \cup S$, computed by Algorithm 2. The graph is constructed from $G_y$ by a number of contractions, which do not alter reachability, i.e. for $s \in S$ the node $\{s\}$ is reachable from the node $Y_v$ containing $v$ in $G_X$ exactly if $s$ is reachable from $v$ in $G_y$. Since all edge weights $w_X$ are strictly positive, in the corresponding Markov chain all transition

---

[14]This class is reminiscent of a class of parametric Markov chains studied by Hahn et al. [2011].

probabilities on the edges of $G_X$ are strictly positive as well. This gives $\{s\}$ a strictly positive absorbing probability when starting a random walk in $Y_v$ exactly if $s$ is reachable from $v$ in $G_y$.

Our next observation is that $\hat{\mathcal{F}} = \mathcal{F} \setminus \{Y \in \mathcal{F} \mid v \in Y\} \cup \{\{v\}\}$, $\hat{y}(\{v\}) = 1$ and $y(Y) = \hat{y}(Y)$ for all $Y \in \hat{\mathcal{F}} \setminus \{\{v\}\}$. Consider the computation of $\mathcal{F}$ in Algorithm 1. Since the output is unique (see Lemma 2 statement (i)), we can assume without loss of generality that after initializing $\mathcal{F}$, all sets in $\{Y \in \mathcal{F} \mid v \notin Y\}$ are added to $\mathcal{F}$ first and then the remaining sets $\{Y \in \mathcal{F} \mid v \in Y\}$. In $\hat{G}$, the only edges missing are the outgoing edges from $v$, therefore, when applying Algorithm 1 to $\hat{G}$ all sets in $\{Y \in \mathcal{F} \mid v \notin Y\}$ can be added to $\hat{\mathcal{F}}$ first (with $\hat{y}(Y) = y(Y)$). Note, that the set $\{v\}$ with $y(\{v\}) = 1$ was added to $\hat{\mathcal{F}}$ in the initialization. We claim, that the algorithm terminates at that point. Suppose not, then there must be another strongly connected component $X \subseteq N$ with $\delta^+(X) \cap E_{\hat{y}} = \emptyset$. If $v \in X$ then since $v$ has no outgoing edges $X = \{v\}$, which is already in $\mathcal{F}$. If $v \notin X$ then $X$ would have already been added.

With these two observations, we can show the following claim: For every casting voter $s \in S \setminus S_v$ the voting weight remains equal, when $v$ turns into a casting voter, i.e., $\pi_s(A) = \pi_s(\hat{A})$. Fix $s \in S \setminus S_v$ and let $U \subset N$ be the set of nodes not reachable from $v$ in $G_y$. We know that $\hat{\mathcal{F}} = \mathcal{F} \setminus \{Y \in \mathcal{F} \mid v \in Y\} \cup \{v\}$, which implies that for every node $u \in U$ the sets containing $u$ are equal in $\mathcal{F}$ and $\hat{\mathcal{F}}$, i.e., $\{Y \in \mathcal{F} \mid u \in Y\} = \{Y \in \hat{\mathcal{F}} \mid u \in Y\}$. Therefore, the outgoing edges from any $u \in U$ are equal in $G_y$ and $\hat{G}_{\hat{y}}$. Since $\hat{\mathcal{F}} \subseteq \mathcal{F}$, the edges in $\hat{G}_{\hat{y}}$ are a subset of the edges in $G_y$ and therefore the set $U$ is not reachable from $v$ in $\hat{G}_{\hat{y}}$. When translating $\hat{G}_{\hat{y}}$ into the Markov chain $(\hat{G}_{\hat{y}}, \hat{P}^{(\varepsilon)})$ (see Equation 1), we get for the probability of any tight out-edge $e$ of $u$ and any $\varepsilon > 0$, that $P_e^{(\varepsilon)} = \hat{P}_e^{(\varepsilon)}$, where $P^{(\varepsilon)}$ is the transition matrix induced by the original graph $G_y$. In the following we argue about the set of walks in $G_y$ and $G_{\hat{y}}$. To this end we define for every $u \in N$, the set $\mathcal{W}[u, s]$ ($\hat{\mathcal{W}}[u, s]$, respectively) as the set of walks in $G_y$ (in $G_{\hat{y}}$, respectively) that start in $u$ and end in sink $s$. Since all walks from any $u \in U$ to $s$ contain only outgoing edges from nodes in $U$, we have $\hat{\mathcal{W}}[u, s] = \mathcal{W}[u, s]$. For any other voter $w \in N \setminus U$ we have $\hat{\mathcal{W}}[w, s] = \mathcal{W}[w, s] = \emptyset$ and therefore

$$\pi_s(\hat{A}) = 1 + \sum_{u \in U} \lim_{\varepsilon \to 0} \sum_{\hat{W} \in \hat{\mathcal{W}}[u,s]} \prod_{e \in \hat{W}} P_e^{(\varepsilon)} = 1 + \sum_{u \in U} \lim_{\varepsilon \to 0} \sum_{W \in \mathcal{W}[u,s]} \prod_{e \in W} P_e^{(\varepsilon)} = \pi_s(A) \quad ,$$

which concludes the proof of the claim.

Summarizing, we know that that for any casting voter $s \in S \setminus S_v$ we have $\pi_s(A) = \pi_s(\hat{A})$, which directly implies that $\sum_{s \in S_v} \pi_s(A) = \pi_v(\hat{A}) + \sum_{s \in S_v} \pi_s(\hat{A})$. $\square$

**Theorem 8 ($\bigstar$).** *The* RANDOM WALK RULE *satisfies confluence.*

*Proof.* Before proving the claim, we introduce notation. For any walk $W$ in some graph $G$, and some node $v \in V(G)$, we define $W[v]$ to be the subwalk of $W$ that starts at the first occasion of $v$ in $W$. For two nodes $u, v \in V(G)$, we define $W[u, v]$ to be the subwalk of $W$ that starts at the first occasion of $u$ and ends at the first occasion of $v$. (Note that $W[v]$ and $W[u, v]$ might be empty.) Now, for a set of walks $\mathcal{W}$ and $u, v, s \in V(G)$, we define $\mathcal{W}[v] = \{W[v] \mid W \in \mathcal{W}\}$ and $\mathcal{W}[u, v] = \{W[u, v] \mid W \in \mathcal{W}\}$. Lastly, we define $\mathcal{W}[u, v, s] = \{W \in W[u, s] \mid v \in W[u, s]\}$. We usually interpret a walk $W$ as a sequence of nodes. In order to facilitate notation, we abuse notation and write $v \in W$ for some node $v \in V(G)$ in order to indicate that $v$ appears in $W$, and for an edge $e \in E(G)$, we write $e \in W$ to indicate that tail and head of $e$ appear consecutively in $W$.

For the remainder of the proof we fix $\mathcal{W}$ to be the set of walks in the input delegation graph $G$ starting in some node from $N$ and ending in some sink node $S$. Moreover, let $G_X$ be the graph at the end of Algorithm 2, i.e., $G_X$ for $X = N \cup S$. We fix $\hat{\mathcal{W}}$ to be the set of walks which start in some node of $G_X$ and end in some sink node of $G_X$ (which are exactly the nodes in $\{\{s\} \mid s \in S\}$).

In the following, we define for every $v \in N$ a probability distribution $f_v : \mathcal{W}[v] \to [0, 1]$, such that it witnesses the fact that the RANDOM WALK RULE is confluent. To this end, we define a mapping $\gamma_v : \hat{\mathcal{W}}[Y_v] \to \mathcal{W}[v]$, where $Y_v$ is the node in $G_X$ that contains $v$. Given a walk $\hat{W} \in \hat{\mathcal{W}}[Y_v]$, we construct $\gamma_v(\hat{W}) \in \mathcal{W}[v]$ as follows: Let $\hat{W} = Y^{(1)}, \ldots Y^{(k)}$. By construction of $G_X$ we know that for every $i \in \{1, \ldots, k\}$, the fact that $(Y^{(i)}, Y^{(i+1)}) \in E_X$ implies that there exists $(b^{(i)}, a^{(i+1)}) \in E$ with $b^{(i)} \in Y^{(i)}$ and $a^{(i+1)} \in Y^{(i+1)}$. Moreover, we define $a^{(1)} = v$ and $b^{(n)} = s$,

where $\{s\} = Y^{(k)}$. Under this construction it holds that $a^{(i)}, b^{(i)} \in Y^{(i)}$ for all $i \in \{1, \ldots, k\}$, but the two nodes may differ. Therefore, we insert subwalks $W^{(i)}$ connecting $a^{(i)}$ to $b^{(i)}$ by using only nodes in $Y^{(i)}$ and visiting each of these nodes at least once. The final walk $\gamma_v(\hat{W})$ is then defined by $(a^{(1)}, W^{(1)}, b^{(1)}, \ldots, a^{(n)}, W^{(n)}, b^{(n)})$. We remark that this mapping is injective, and it holds that $\hat{W}$ visits some node $Y \in V(G_X)$ if and only if $\gamma_v(\hat{W})$ visits all nodes in $Y$.

Recall that the assignment $A$ of the RANDOM WALK RULE can be computed via a Markov chain $(G'_X, P)$ derived from the contracted graph $(G_X, w_X)$ (see Section 5 and Section 6), where $G'_X$ is derived from $G_X$ by adding self-loops. In Lemma A.1 we show that introducing (and thus removing) self-loops to states in an absorbing Markov chain does not change its absorbing probabilities. We retrieve the Markov chain $(G_X, \hat{P})$ by removing all self loops of all voters in $N$ and rescaling the other probabilities accordingly. We then make use of this Markov chain in order to define $f_v$ over $\mathcal{W}[v]$. That is, for any $W \in \mathcal{W}[v]$ let

$$f_v(W) = \begin{cases} \prod_{e \in \hat{W}} \hat{P}_e & \text{if there exists } \hat{W} \in \hat{\mathcal{W}}[Y_v] \text{ such that } \gamma_v(\hat{W}) = W \\ 0 & \text{else.} \end{cases}$$

Note that, the above expression is well defined since $\gamma_v$ is injective.

In the remainder of the proof, we show that $f_v$ witnesses the confluence of the RANDOM WALK RULE. First, we show that $f_v$ is indeed consistent with the assignment $A$ returned by RANDOM WALK RULE. That is, for any $v \in N$ and $s \in S$ it holds that

$$\mathbb{P}_{W \sim f_v}[s \in W] = \sum_{W \in \mathcal{W}[v,s]} f_v(W) = \sum_{\hat{W} \in \hat{\mathcal{W}}[Y_v, \{s\}]} \prod_{e \in \hat{W}} \hat{P}_e = A_{v,s} \quad .$$

The second equality comes from the fact that $\gamma_v$ is injective and exactly those walks in $\hat{\mathcal{W}}[Y_v, \{s\}]$ are mapped by $\gamma_v$ to walks in $\mathcal{W}[v,s]$. Moreover, all walks in $\mathcal{W}[v,s]$ that have no preimage in $\hat{\mathcal{W}}[Y_v, \{s\}]$ are zero-valued by $f_v$. The last equality comes from the fact that $A_{v,s}$ equals the probability that the Markov chain $(G'_X, P)$ (equivalently, $(G_X, \hat{P})$) reaches $\{s\}$ if started in $Y_v$ (see Section 5 and Section 6).

We now turn to the second condition on the family of probability distributions $f_v, v \in N$. That is, for every $u, v \in N, s \in S$ it holds that

$$\mathbb{P}_{W \sim f_u}[v \in W \wedge s \in W] = \sum_{W \in \mathcal{W}[u,v,s]} f_u(W) = \sum_{\hat{W} \in \hat{\mathcal{W}}[Y_u, Y_v, \{s\}]} \prod_{e \in \hat{W}} \hat{P}_e$$

$$= \sum_{\hat{W} \in \hat{\mathcal{W}}[Y_u, Y_v, \{s\}]} \left( \prod_{e \in \hat{W}[Y_u, Y_v]} \hat{P}_e \right) \left( \prod_{e \in \hat{W}[Y_v, \{s\}]} \hat{P}_e \right)$$

$$= \left( \sum_{\hat{W} \in \hat{\mathcal{W}}[Y_u, Y_v]} \prod_{e \in \hat{W}} \hat{P}_e \right) \cdot \left( \sum_{\hat{W} \in \hat{\mathcal{W}}[Y_v, \{s\}]} \prod_{e \in \hat{W}} \hat{P}_e \right)$$

$$= \left( \sum_{s' \in S} \sum_{\hat{W} \in \hat{\mathcal{W}}[Y_u, Y_v, \{s'\}]} \prod_{e \in \hat{W}} \hat{P}_e \right) \cdot \left( \sum_{\hat{W} \in \hat{\mathcal{W}}[Y_v, \{s\}]} \prod_{e \in \hat{W}} \hat{P}_e \right)$$

$$= \left( \sum_{s' \in S} \sum_{W \in \mathcal{W}[u,v,s']} f_u(W) \cdot \left( \sum_{W \in \mathcal{W}[v,s]} f_v(W) \right) \right)$$

$$= \mathbb{P}_{W \sim f_u}[v \in W] \cdot \mathbb{P}_{W \sim f_v}[s \in W].$$

The second equality follows from the same reason as above, i.e., $\gamma_v$ is injective, exactly those walks in $\hat{\mathcal{W}}[Y_u, Y_v, \{s\}]$ are mapped by $\gamma_v$ to walks in $\mathcal{W}[u,v,s]$, and all walks in $\mathcal{W}[u,v,s]$ that have no preimage in $\hat{\mathcal{W}}[Y_u, Y_v, \{s\}]$ are zero-valued by $f_v$. The third inequality holds by the fact that every walk that is considered in the sum can be partitioned into $\hat{W}[Y_u, Y_v]$ and $\hat{W}[Y_v, \{s\}]$. The fourth equality follows from factoring out by the subwalks. The fifth equality follows from the fact that every walk in $\hat{\mathcal{W}}$ reaches some sink node eventually, and therefore, the additional factor in the first bracket sums up to one. Lastly, the sixth equality follows from the very same argument as before.

From the above equation we get in particular that for every $u, v \in N, s \in S$ it holds that

$$\mathbb{P}_{W \sim f_u}[s \in W \mid v \in W] = \frac{\mathbb{P}_{W \sim f_u}[s \in W \wedge v \in W]}{\mathbb{P}_{W \sim f_u}[v \in W]} = \mathbb{P}_{W \sim f_v}[s \in W].$$

This concludes the proof. □

The next axiom was in its essence first introduced by Behrens and Swierczek [2015] and first given the name *guru-participation* in Kotsialou and Riley [2020]. The idea is that a representative (the *guru*) of a voter, should not be worse off if said voter abstains from the vote. Brill et al. [2022] define this property for non-fractional ranked delegations by requiring that any casting voter that was not a representative of the newly abstaining voter should not loose voting weight. This definition translates well into the setting of fractional delegations where we can have multiple representatives per voter. For simplicity, we made a slight modification to the definition[15], resulting in a slightly stronger axiom.

Previously, we stated the general assumption that every delegating voter in a delegation graph $(G, c)$ has a path to some casting voter in $G$. In this section we modify given delegation graphs by removing nodes or edges, which may result in an invalid delegation graph not satisfying this assumption. To prevent this, we implicitly assume that after modifying a delegation graph, all nodes in $N$ not connected to any sink in $S$ (we call them *isolated*) are removed from the graph.

**Guru Participation:** A delegation rule satisfies *guru-participation* if the following holds for every instance $(G, c)$: Let $(\hat{G}, c)$ be the instance derived from $(G, c)$ by removing a node $v \in N$ (and all newly isolated nodes), let $S_v = \{s \in S \mid A_{v,s} > 0\}$ be the set of representatives of $v$ and let $A$ and $\hat{A}$ be the assignments returned by the delegation rule for $(G, c)$ and $(\hat{G}, c)$, respectively. Then

$$\pi_s(\hat{A}) \geq \pi_s(A) \quad \forall s \in S \backslash S_v \quad .$$

In particular, this implies

$$\sum_{s \in S_v} \pi_s(\hat{A}) + 1 \leq \sum_{s \in S_v} \pi_s(A) \quad .$$

In order to prove that the RANDOM WALK RULE satisfies guru-participation we first show the following lemma, saying that the voting weight of no casting voter decreases, when the in-edges of another casting voter are removed from the graph.

**Lemma C.1.** *For the* RANDOM WALK RULE*, removing the incoming edges of some casting voter $s \in S$ (and all newly isolated voters) does not decrease the absolute voting weight of any casting voter $s' \in S \setminus \{s\}$.*

*Proof.* Let $(G, c)$ be a delegation graph and $s \in S$ a sink. Let $(\hat{G}, c)$ be the delegation graph, where the in-edges of $s$ and all voters disconnected from casting voters are removed. Let $P^{(\varepsilon)}$ and $\hat{P}^{(\varepsilon)}$ be the transition matrices of the corresponding Markov chains $M_\varepsilon$ and $\hat{M}_\varepsilon$. Then, for any $\varepsilon > 0$ and edge $e$ in $\hat{G}$ we have $P_e^{(\varepsilon)} \leq \hat{P}_e^{(\varepsilon)}$. Since no edge on a path from any $v \in N$ to any $s' \in S \setminus \{s\}$ was removed, we have $\hat{\mathcal{W}}[v, s'] = \mathcal{W}[v, s']$ and $\hat{P}_e^{(\varepsilon)} \geq P_e^{(\varepsilon)}$ for every edge $e$ in $\hat{G}$ and $\varepsilon > 0$. Therefore, for the absolute voting weight of any $s' \in S \setminus \{s\}$ in $\hat{G}$ we get

$$\pi_{s'}(\hat{A}) = 1 + \sum_{v \in N} \lim_{\varepsilon \to 0} \sum_{\hat{W} \in \hat{\mathcal{W}}[v, s']} \prod_{e \in \hat{W}} P_e^{(\varepsilon)} \geq 1 + \sum_{v \in N} \lim_{\varepsilon \to 0} \sum_{W \in \mathcal{W}[v, s']} \prod_{e \in W} P_e^{(\varepsilon)} = \pi_{s'}(A) \quad ,$$

which concludes the proof. □

Using Lemma C.1 and the proof of Theorem 7, we can show that guru-participation is satisfied by the RANDOM WALK RULE by removing a delegating voter step by step.

**Theorem C.2.** *The* RANDOM WALK RULE *satisfies guru participation.*

*Proof.* Let $(G, c)$ be a delegation graph and $v \in N$ a delegating voter. We remove $v$ from $G$ in three steps. First, we remove all out-edges of $v$, making $v$ a casting voter and call the new delegation graph $(\hat{G}_1, c)$. Then we remove the in-edges of $v$ (and all newly isolated voters) and get $(\hat{G}_2, c)$. Finally, we remove $v$ itself to retrieve $(\hat{G}, c)$ as in the definition of guru-participation. Let $A$, $\hat{A}_1$, $\hat{A}_2$ and $\hat{A}$ be the assignments returned by the RANDOM WALK RULE for $(g, c)$, $(\hat{G}_1, c)$, $(\hat{G}_2, c)$ and

---

[15]More specifically, Brill et al. [2022] use the notion of *relative* voting weight between the casting voters in the definition of the axiom, which follows from our version of the axiom using absolute voting weight.

$(\hat{G}, c)$, respectively. From the proof of Theorem 7 we know that for every casting voter $s \in S \setminus S_v$ the voting weight in the instances $(G, c)$ and $(\hat{G}_1, c)$ is equal, i.e., $\pi_s(\hat{A}_1) = \pi_s(A)$. From Lemma C.1 it follows that the voting weight of these voters can only increase if also the in-edges of $v$ are removed, i.e., $\pi_s(\hat{A}_2) \geq \pi_s(\hat{A}_1)$. Finally, removing the now completely isolated (now casting) voter $v$ does not change the absolute voting weight of any other voter and therefore $\pi_s(\hat{A}) \geq \pi_s(A)$. □

## C.1 Relation to the Axioms of Brill et al. [2022]

First, we remark that the definition of a non-fractional delegation rule varies slightly from the definition of a delegation rule in Brill et al. [2022]. That is, Brill et al. [2022] define the output of a delegation rule as a mapping from each delegating voter to some *path* to a casting voter. Here, on the other hand, we define the output of a non-fractional delegation rule as a (non-fractional) assignment of delegating voters to casting voters. Hence, the definition of a delegation rule by Brill et al. [2022] is slightly more restrictive than our definition, hence, ceteris paribus, the impossibility result holds in particular for the smaller set of delegation rules. In the following, we refer to our definition as non-fractional delegation rules and to the definition of Brill et al. [2022] as non-fractional* delegation rules. We say that a non-fractional delegation rule is *consistent* to a non-fractional* delegation rule if, for any input, the assignment in the former corresponds to the induced assignment in the latter.

**Copy-robustness** Next, consider the copy-robustness axioms. The axiom by Brill et al. [2022] for non-fractional* delegation rules differs to our copy-robustness axiom restricted to non-fractional delegation rules in two technicalities: First, Brill et al. [2022] consider the relative voting weight instead of the absolute voting weight. However, since the number of voters does not change from $(G, c)$ to $(\hat{G}, c)$, this does not change the axiom. The other difference is that their axiom requires that the delegating voter $v$ under consideration has a direct path (in the output of the non-fractional* delegation rule) to its assigned casting voter. Since a non-fractional delegation rule only outputs an assignment and no path, we relaxed this assumption. Hence, our copy-robustness axiom is slightly stronger than the one presented by Brill et al. [2022]. Nevertheless, it is easy to see that also the weaker version of the axiom is necessarily violated within the proof of the impossibility theorem when we utilize the definition of a delegation rule by Brill et al. [2022].

**Confluence** Lastly, consider the confluence axiom. Brill et al. [2022] define their confluence axiom, which we denote by *confluence** as follows: A non-fractional* delegation rule satisfies confluence* if, for every delegating voter $v \in N$ exactly one outgoing edge of $v$ appears within the union of paths returned by the delegation rule. In particular, this is equivalent to the fact that the union of the returned paths forms a branching in the delegation graph. We prove below that the two axioms are in fact equivalent (within the restricted domain of non-fractional delegation rules).

**Proposition 1.** *A non-fractional* delegation rule satisfies confluence* if and only if there exists a consistent non-fractional delegation rule that satisfies confluence.*

*Proof.* We start by proving the forward direction. Consider a non-fractional* delegation rule that satisfies confluence*. We define a consistent non-fractional delegation rule by simply returning the assignment induced by the returned paths instead of the paths. For showing that the rule satisfies confluence, we define the probability distributions $f_v, v \in N$ by setting $f_v(W) = 1$ if and only if $W$ is the path returned for $v$ by the delegation rule. All other probabilities are set to zero. Then, confluence* directly implies that the distributions $f_v, v \in N$ witness confluence.

For the other direction, consider a non-fractional delegation rule satisfying confluence, and let $f_v, v \in N$ be the probability distributions that witness this fact. Building upon that, we define a branching in $G$ that is consistent with the outcome of the delegation rule. Interpreting this branching as the output of a non-fractional* delegation rule then proves the claim. We construct the branching as follows: We first set $B = \emptyset$. In the beginning, set all delegating voters to be "active", while all casting voters are "inactive". Now, pick some arbitrary active voter $v \in N$ and consider some arbitrary walk $W$ that obtains non-zero probability by the distribution $f_v$. Construct a path $P$ from $W$ by cutting the walk in the first appearance of some inactive voter, and then short cutting all remaining cycles (if existent). Now, add all edges in $P$ to $B$ and set all delegating voters on $P$ to be "inactive". Note that, by confluence, for each newly inactive voter $u \in N$ it holds that the casting voter assigned by the delegation rule corresponds to the sink node at the end of the unique maximal path in $B$ starting

from $u$. We continue this process until all voters are inactive. As a result, we created a branching that is consistent with the original delegation rule, hence, there exists a consistent non-fractional* delegation rule that implicitly returns branchings, i.e., is confluent. $\qquad\square$

## D  Broader Impact

We are aware of the fact that any delegation rule, and in particular the one suggested in this paper, may be implemented in a liquid democracy system and could thereby have real world impact. In this paper, we chose the axiomatic method in order to evaluate the suggested rule in a principled way. While, with respect to the axioms considered in the literature so far, our delegation rule performs very well, we want to point out that this is the very first paper introducing fractional delegation rules for ranked delegations. In particular, there is a risk of some unforeseen disadvantages of the rule that could possibly be used for manipulations or lead to other negative societal effects. Therefore, we think that further theoretical and also empirical research is necessary before recommending our suggested delegation rule for (high-stake) real-world decision making.

