741 **Top-rank priority:** For any delegation graph and output of the delegation rule $A$, if voter $v \in N$ has
742 exactly one outgoing edge of cost 1 and that edge ends in a casting voter $s \in S$, then $A_{v,s} = 1$.

743 **Theorem C.3.** MIXED BORDA BRANCHING *satisfies top-rank priority.*

744 *Proof.* Let $G$, $v$ and $s$ be defined as above. We show that for the assignment returned by MIXED
745 BORDA BRANCHING $A_{v,s} = 1$ by showing that every Borda branching contains the edge $(v, s)$.
746 Suppose there is a Borda branching $B'$ with $(v, s) \notin B$, then we construct a new branching $\hat{B}$ by
747 removing the out-edge of $v$ from $B'$ and adding $(v, s)$ instead. $\hat{B}$ is a branching, since no cycles can
748 be introduced by adding an edge to a sink and $|\hat{B}| = |B'|$. Since $v$ has only one outgoing edge of cost
749 one, $\hat{B}$ has lower total cost that $B'$, contradicting the assumption that $B'$ is a Borda branching. $\quad\square$

# D  Broader Impact

751 We are aware of the fact that any delegation rule, and in particular the one suggested in this paper,
752 may be implemented in a liquid democracy system and could thereby have real world impact. In this
753 paper, we chose the axiomatic method in order to evaluate the suggested rule in a principled way.
754 While, with respect to the axioms considered in the literature so far, our delegation rule performs
755 very well, we want to point out that this is the very first paper introducing fractional delegation rules
756 for ranked delegations. In particular, there is a risk of some unforeseen disadvantages of the rule
757 that could possibly be used for manipulations or lead to other negative societal effects. Therefore,
758 we think that further theoretical and also empirical research is necessary before recommending our
759 suggested delegation rule for (high-stake) real-world decision making.