# OpenReview forum: "Anonymous and Copy-Robust Delegations for Liquid Democracy"
_NeurIPS.cc/2023/Conference — NeurIPS 2023 spotlight_

### Official Review · Reviewer_PkVV · 2023-07-04

**Soundness:** 4 excellent
**Presentation:** 4 excellent
**Contribution:** 3 good
**Rating:** 7
**Confidence:** 4

**Summary:**

The paper studies fractional allocation rules when each agent indicates a ranking over the agents that agrees to represent her, to overcome impossibility results of deterministic rules. The authors consider two different rules which are shown that are equivalent. They also provide a polynomial time algorithm for finding the outcome of one of the two rules.

**Strengths:**

-Very well-written paper

-Interesting model

-Natural extension of the axioms to fractional allocations

-Technically involved and interesting results

-Use of known lemmas in interesting and elegant ways

Further notes:

Line 73: "casting voter"

Line 134: I think you mean "|B|=|V(G)|-1"


**Weaknesses:**

The choice of  the top-rank priority axiom, while it is quite natural, seems a bit arbitrary. It would be nice to show what kind of other axioms would lead to the same impossibility result.

**Questions:**

See weaknesses above

**Limitations:**

 The authors adequately addressed the limitations of their work.

---

> ### Author Rebuttal · Authors · 2023-08-09
>
> The choice of top-rank priority is motivated by the proof of the impossibility theorem for the non-fractional setting (Section 1). Consider the example instance with one delegating voter $v_1$ and three casting voters $s_1$, $s_2$, $s_3$, where $v_1$'s first delegation preference is $s_1$ and her second (and last) delegation preference is $s_2$. We show that any anonymous, confluent and copy-robust rule must assign $v_1$'s weight to $s_2$, which contradicts the idea of delegation preferences. The following versions of a fourth property can conclude the impossibility result, without changing any of the results of our paper.
>
> (i) In a situation where a voter $v$ has exactly one outgoing edge of rank one and exactly one of rank two, leading directly to casting voters $s_1$ and $s_2$, respectively, a delegation rule should assign all of $v$'s voting weight to $s_1$.\
> (ii) In a situation where $v$ has exactly one outgoing edge of rank one and it leads directly to a casting voter $s_1$, a delegation rule should assign all of $v$'s voting weight to $s_1$.\
> (iii) A delegation rule should distribute $v$'s voting weight only over casting voters reachable via a walk through the graph that starts with a rank one edge.
>
> While (i) is extremely specific to the situation in the proof of the impossibility result, (iii) is relatively general. At the same time it is debatable whether (iii) is a desired property, while this is quite clear for (i). Due to this trade-off between arbitrariness and certainty of desiredness we choose (ii) as the definition of top-ranked priority, being slightly more general than (i) while still certainly being desirable.

---

> > ### Comment · Reviewer_PkVV · 2023-08-18
> >
> > Thanks for your response. I don't have other questions.

---

### Official Review · Reviewer_rLuU · 2023-07-06

**Soundness:** 4 excellent
**Presentation:** 4 excellent
**Contribution:** 4 excellent
**Rating:** 8
**Confidence:** 3

**Summary:**

This paper primarily examines two equivalent fractional delegation rules for Liquid Democracy (transitive delegate voting): Mixed Borda Branching and Random Walk Rule. The main contribution of this paper is the equivalence between these two rules in the generalized setting of fractional delegations. This result is complemented by an axiomatic analysis of the rules w.r.t. notions of anonymity, copy-robustness, confluence, and top-rank priority.

**Strengths:**

The equivalence between the two delegation rules is non-obvious and a significant contribution. Moreover, the proof of equivalence is highly non-trivial. These results in Sections 4-5 are of considerable value. But I am not sure if those alone are enough to carry the paper (see Weaknesses). Although perhaps they are.

**Weaknesses:**

Unless I have misunderstood something, it doesn’t appear that the axiomatic analysis really solves, or resolves, any of the important issues discussed in the introduction. It circumvents them instead in a way that feels cheap.


First, allowing fractional delegations clearly means confluence is no longer a constraint. The authors state essentially this in an incredibly oblique way in the proof of Theorem 8, “One can verify that every delegation rule that can be formalized via a Markov chain on the delegation graph (G,c) satisfies confluence.” Confluence requires that if v1 delegates to v2 that the remainder of their delegation path much be consistent. Suppose, for example, that v1 is the only voter delegating to v2, and v2’s delegation is split between v3 and v4. No we “send” ½ of v1’s vote and ½ of v2’s vote to v3, and the remaining ½ + ½ to v4. Of course, this is functionally equivalent to sending v1’s entire vote to v3 and v2’s entire vote to v4, violating confluence. In reality, the axiom of confluence wasn’t generalized, it is made irrelevant, because any delegation that violates confluence can be trivially converted to a fractional one that satisfies its relaxation. Whether we talk in terms of dividing vote tokens or probability distributions over delegations doesn’t matter, it’s the same principle. Similarly, simple path vs. walk makes no difference here.

Second, the authors state that the purpose of giving delegations as orderings is to prevent delegation cycles and isolated voters. In footnote 3 on page 4, the authors admit that they handle isolated voters by ignoring them entirely and removing them from the graph. However it is not discussed how this bears on the axioms. Ignoring such voters is the same as assigning them a voting weight of zero, whereas if they cast their vote directly they get a voting weight of at least one (and possibly more if other isolated voters delegate to them). This violates copy-robustness, but this violation does not appear to be mentioned.


Lastly, while this is a minor note, the authors should not say that they generalized the axioms, but rather that they relaxed the axioms. The authors generalized the class of delegation rules, and relaxed the axioms required.

Edit:
Based on the author(s) rebuttal, I have improved my understanding of the paper and its results.I no longer stand by my original criticism.

**Questions:**

Q1: Am I correct in saying that ignoring the isolated voters ultimately violates copy-robustness?

Q2: Am I correct that fractional delegation trivializes the axiom of confluence?

**Limitations:**

No further limitations.

---

> ### Author Rebuttal · Authors · 2023-08-09
>
> **Q1**: Am I correct in saying that ignoring the isolated voters ultimately violates copy-robustness?\
> **A:** No.\
> Before justifying our answer, we want to clarify the handling of isolated voters. The motivation for introducing ranked delegations is to mitigate the risk of having isolated voters, i.e., voters that cannot reach any casting voter through a chain of delegations. Of course even with ranked delegations there can still be isolated voters, however it was empirically shown (Brill et al. [2022]) that few backup delegations lead to almost no isolated voters in many random graph models. We remove remaining isolated voters from the instance as is standard in the liquid democracy literature, since there is no way of assigning them reasonable representatives. We can then assume for all our definitions that each delegating voter has a path to a casting voter in the delegation graph.
>
> That said, the axiom copy-robustness captures the impossibility of manipulation by a delegating voter copying the vote of its representative(s). Assuming, this delegating voter knows its representative(s) chosen by some delegation rule, the voter could copy their vote instead of delegating. Copy-robustness then requires that this does not change the joint voting weight of the delegating voter and its representative(s), thus preventing manipulations of this type.
>
> Since an isolated voter has no representatives and therefore nobodies vote to copy, the described situation of an isolated voter deciding to being a casting voter instead does not classify as a manipulation and is therefore not (and should not be) captured by the copy-robustness axiom. In fact, in a model that includes isolated voters, an isolated voter increasing its voting weight by casting its vote would be expected (and desired) behavior. \
>
> **Q2**: Am I correct that fractional delegation trivializes the axiom of confluence?\
> **A:** No.\
> Before justifying our answer, we would like to emphasize that a delegation rule **assigns each delegating voter** a probability distribution over casting voters. Crucially, confluence captures a consistency across the distributions assigned to different casting voters. \
> Considering only the total voting weight of v3 and v4 in the example given by the reviewer and calling all delegation rules yielding this outcome 'functionally equivalent' contradicts our central definition of a delegation rule and also the idea of confluence to compare the voters individual assignments. This means that even though the two example assignments yield the same outcome in terms of total voting weight, the underlying delegation rules are different and can have different axiomatic properties.
>
> To show the non-trivial nature of confluence on an example, assume that in the example given by the reviewer, voter v2's first preference for delegation is v3 and v4 is the second. In this extended example, top-rank priority alone would only require v2's vote to be delegated fully to v3. However, if we enforce confluence as well, then v1's vote must be delegated to v3 as well.
>
> **C:** Lastly, while this is a minor note, the authors should not say that they generalized the axioms, but rather that they relaxed the axioms. The authors generalized the class of delegation rules, and relaxed the axioms required.\
> **A:** We would like to clarify exactly what we mean by 'generalizing' an axiom. The axioms we examined were defined on a model without fractional delegations, which we generalized to a model with fractional delegations, as recognized by the reviewer. Some of the axioms then needed adjustment to this new setting. We adjusted the axioms in a (to our perception) natural way, making sure that if applied to the special case of a non-fractional delegation rule, they correspond precisely to the original axioms. (For all axioms besides confluence this fact is easy to see. For confluence, we recently wrote a proof for this claim, which we are happy to provide upon request.) We therefore refer to the new axioms as 'generalized' in contrast to 'relaxed' which is the term we would use for an axiom that was weakened, i.e. that is implied by its original version.

---

> > ### Comment · Reviewer_rLuU · 2023-08-16
> > **Updating my View**
> >
> > First, I thank the authors for their detailed answers to my questions.
> >
> > I can see now that view that my criticism that fractional delegation trivializes the axiom of confluence was incorrect. I retract this criticism.
> >
> > It still appears there is a small problem with isolated voters, but this is an inescapable part of the nature of the problem. There is, in a sense, a violation of the axiom, but not in a way that conflicts with the motivation, and this is certainly not a reason for rejection.
> >
> > My view of the paper is now strongly positive and I argue for acceptance.

---

### Official Review · Reviewer_eBke · 2023-07-06

**Soundness:** 4 excellent
**Presentation:** 4 excellent
**Contribution:** 3 good
**Rating:** 7
**Confidence:** 4

**Summary:**

The authors study liquid democracy with fractional (i.e., splittable) delegations. They extend previous work by Brill et al. on delegation rules that satisfy anonymity and copy-robustness, and demonstrate that two delegation rules (mixed Borda branching and the random walk rule) that were previously thought to be different and each satisfy one property are actually the same rule that satisfies (generalized versions of) both properties. Their algorithm for computing the outcome of the combined rule is also the first efficient algorithm for a problem in semi-supervised learning, the directed power watershed.

**Strengths:**

+ The paper is well-written and easy to follow, and the problem they study is well-motivated in the context of liquid democracy.
+ The algorithm is nontrivial and of independent interest to other communities in computer science.
+ It's also nice to see that two previously-proposed rules are actually the same; it's extra satisfying that this rule happens to satisfy generalizations of two (really, four) axioms that were thought to be hard to simultaneously satisfy.

**Weaknesses:**

- My main hesitation with this paper is the use of fractional delegations in liquid democracy. One central tenet of LD is the ability to immediately remove a delegation from someone who cast a vote in a way you did not approve of, which becomes much more difficult with splittable delegations. Additionally, if you allow agents to know exactly where portions of their vote ended up, they will potentially be given a lot of information about the rest of the delegation network.
- Typos: line 73: casting, line 166: copy-robust, not copy-robustness, line 342: lens

**Questions:**

1. Have you thought about ways of explaining the outcomes of MBB / RWR to users of liquid democracy systems? Are there intuitive ways of showing the flow of splittable votes through the network?

**Limitations:**

Yes

---

> ### Author Rebuttal · Authors · 2023-08-09
>
> A possible explanation could be the following reinterpretation of the rules, motivated by the nature of the algorithm. In the last step of the algorithm we solve an absorbing Markov chain on a partially contracted delegation graph, where the outgoing edge probabilities of each contracted vertex depend on its inner structure of delegation preferences. The assignment of the rule will then be exactly the same for any voters contained in the same contracted vertex. The rule could therefore be interpreted (and explained) as a rule aggregating the lower delegation preferences of voter clusters that only delegate to one another with their higher preferences. We could then present the flow constructed from the Markov chain on the contracted graph as an explanation on how the aggregated delegation preferences are finally resolved into delegations. This flow is polynomial time computable, since it is a byproduct of the computation of the absorbing probabilities. A downside of this method is that it only offers a partial explanation since the aggregation of delegation preferences is not explained in detail.
>
> If we want to construct a flow on the (uncontracted) delegation graph directly, there are multiple natural ways to do this for each delegating voters vote independently, depending whether we look at the rule from the mixed Borda branching or random walk perspective. However, these individual flows may be inconsistent with another, in the sense that the relative split of flow in a specific voter might be different in the flows of different voters. In fact, we can show that it is in general not possible to construct consistent flows that correspond to the assignment of our rule for all voters.

---

### Official Review · Reviewer_rGK4 · 2023-07-07

**Soundness:** 3 good
**Presentation:** 3 good
**Contribution:** 3 good
**Rating:** 6
**Confidence:** 3

**Summary:**

The paper studies liquid democracy where voters can delegate their votes to others instead of casting them directly. Within this framework, some voters act as casting voters, while others delegate their votes. Delegation rules determine how casting voters are chosen for each set of delegating voters. However, voters have preferences regarding whom they trust more to represent their votes.
When the entire vote must be delegated, it becomes challenging to satisfy all of several desirable axioms simultaneously. Instead, the authors explore fractional/ probabilistic delegations, allowing a vote to be split across casting voters. They demonstrate that by employing the random walk rule, it is possible to recover the possibility of meeting all axioms.
Moreover, the authors note that the distribution over branchings, ensuring that every delegating voter is connected to a casting voter, follows a uniform distribution over all minimum-cost branchings. This connection has also been previously observed in the work of Fita Sanmartin et al.
To compute the delegation distribution efficiently, the authors propose a polynomial time algorithm that builds upon Fulkerson's algorithm.

**Strengths:**

- Fractional delegations are a natural extension and furthermore they are transparent in the sense that a voter can know what proportion of her vote was transferred to which casting voter.
- The result on the possibility of confluence, anonymity and copy-robustness if we move to the realm of fractional delegations is nice to have.
- The authors leverage results from graph theory and combinatorial optimization to obtain their results, hence introducing techniques from this literature to their community.

**Weaknesses:**

- Equivalence of Mixed Borda Branching and Random Walk rule: Given an understanding of the Markov chain rule, the equivalence between the Random Walk rule and Mixed Borda Branching seems immediate. The authors' claim of surprise at this equivalence is somewhat perplexing, as the connection should have been anticipated with knowledge of the Markov chain rule. It would have been more understandable if other authors had defined Mixed Borda without recognizing this link, but the authors themselves acknowledge that this interpretation has already been observed by Rita Sanmartin et al. Consequently, the discussion on equivalence could be shortened by referring to Rita Sanmartin et al.'s work rather than presenting it as a surprising result.
- Algorithm 2 and Directed Powershed: Unfortunately, I cannot confirm whether the Directed Powershed algorithm is novel to this paper. It appears that Sanmartin et al. discuss an extension of the undirected version to address the directed case (Page 5: "In section 5, we show how the Power Watershed can be generalized to directed graphs by means of the DProbWS.”, Page 8: "In the ProbWS paper [12], it was proven that the Power Watershed [6] is equivalent to applying the ProbWS restricted to the minimum cost spanning forests. This restriction corresponds to the case of a Gibbs distribution of minimal entropy over the forests. In this section, we will prove the analogous result for the DProbWS: When the entropy of the Gibbs distribution over the directed in-forests (3.1) is minimal, then DProbWS is restricted to the minimum cost spanning in-forests (mSF). This permits us to define a natural extension of the Power Watershed to directed graphs.
“). So they claim to leverage existing knowledge (Power Watershed: A Unifying Graph-Based Optimization Framework, 2010).  However, if the problem remains unsolved, a discussion of the Power Watershed algorithm and its limitations in extending to the directed case would be necessary. It is worth noting that Algorithm 2 has limitations in practicality, with a runtime of O(n^7) (up to log factors), while Power Watershed may offer better efficiency.
- The anonymity gained through randomizing over mincost branchings seems not surprising,. It is unclear where the challenges lie in proving that the other properties still hold ( as the authors seem to demonstrate more general versions of the axioms).

It would be valuable to have an in-depth discussion of the supervised learning literature, as the problem of liquid democracy appears to be studied under different names with distinct requirements in various fields. Exploring the Power Watershed algorithm and its directed extension, along with providing a comprehensive explanation of the connections between different fields, would be a very useful contribution. This could introduce the supervised learning literature to the computational social choice community, while also allowing the supervised learning community to benefit from the axiomatic analysis conducted in this work. If the extension of power watershed algorithm to the directed from the literature turns out to be faulty or very unclear, I will increase my score.

**Questions:**

- Can you provide an explanation for why Fita Sanmartin et al.'s claim about extending the Power watershed algorithm is incorrect or why it cannot be straightforwardly extended? How does the runtime of your algorithm compare to that of Power Watershed?
- What makes the Equivalence of Mixed Borda and Random Walk Rule surprising?

**Limitations:**

No concerns, limitations are addressed in the appendix.

---

> ### Author Rebuttal · Authors · 2023-08-09
>
> If accepted, we are happy to use the additional page to elaborate on the connection to semisupervised learning, in particular, by providing an in-depth discussion of [1] and [2].
> ## Q1
> **Short** Indeed, we think that some formulations in [1] are ambigious and give the impression that the paper would introduce an **algorithm** for finding the directed Power Watershed solution. Instead, the paper presents an algorithm (DProbWS), which is only well-defined for a fixed parameter $\mu$ and shows that solving a parameterized version of DProbWS and then taking $\mu \rightarrow \infty$ would lead to the directed Power Watershed solution. However, the question whether a parameterized version of DProbWS can be solved efficiently is completely left open. While [2] presents a similar result in the undirected case, they also present an efficient algorithm for the limit case. Thus, when [1] mention a "generalization" to directed graphs, we understand that this only refers to the former result but not to the presentation of an algorithm solving the limit case.
>
> **Detailed** The paper introduces a digraph $G=(U \cup S,E)$ with a cost function $c$ on the edges and weight function: $$w(e) = e^{-\mu c(e)}.$$For every $q \in U,s \in S$, the algorithm solves the linear system $L \cdot x_q^s = -B_s$, and outputs $x_q^{s}$. Importantly, the matrix $L$ and the vector $B_s$ consist only of sums of $w(e), e \in E$. Hence, when $\mu \rightarrow \infty$, all elements of $L$ and $B_s$ may be zero (e.g., if all costs are positive), and the linear system may have infinitely many solutions. In Rem. 1 of [1], the authors acknowledge that the sum of weights of all branchings connecting any $q \in U$ to $S$ needs to be non-zero in order for the algorithm to be well-defined. This is violated for $\mu \rightarrow \infty$.
> We cite Thm 5.1. from [1] in our words: If $\mu \rightarrow \infty$, then $$x_q^{s} = \frac{\text{no.min-cost bran.connecting $q$  to $s$}}{ \text{no.min-cost bran.}},$$ where $x_q^s$ is defined by DProbWS. Given that $x_q^{s}$ is not defined for $\mu \rightarrow \infty$, we believe a more accurate variant of the statement would interpret $x_q^{s}$ as a function of $\mu$ and state $$\lim_{\mu \rightarrow \infty} x_q^{s}(\mu) = \frac{\text{no.min-cost bran. con. $q$ to $s$}}{\text{no.min-cost bran.}}.$$
> This is also shown in the proof.
>
> While DProbWS can compute $x_q^{s_1}(\mu)$ for any $\mu \in \mathbb{R}$, its running time increases in $\mu$, as the running time of solving a linear system depends on the size of the input. Alternatively, one could compute the function $x_q^{s_1}(\mu)$ and then take its limit. We thought along these lines, and tried to build upon alg. for parameterized Markov chains, however, all of the alg. that we found (e.g. [3]) have exponential running time.
>
> ## Q2
>
> **Short** While the power watershed alg. [2] (PW) is similar in spirit to our algorithm, the fact that we consider directed trees with dedicated root nodes leads to a more complex algorithm with a higher running time. We believe that this complexity is inherent to the problem and, to some extent, unavoidable (see detailed answer). We don't find this surprising: Even the classic min-cost spanning tree problem can be solved by a greedy algorithm and forms a Matroid, but this structure gets lost when moving to its directed variant.
>
> **Detailed** Recall the goal of PW (our Alg. 2, respectively). Given an undirected (directed, resp.) graph $G=(N \cup S, E)$ with cost function $c: E \rightarrow \mathbb{N}$, compute for every $v \in N$, $s \in S$, the value $x_{v,s}$ corresponding to the relative number of min-cost spanning forests aka MSF (min-cost branchings, resp.) in which $v$ reaches $s$. To illustrate the different complexities, we restrict ourselves to the special case when the subgraph induced by $N$, i.e., $G[N]$, contains one connected component (strongly con. comp., resp.), edges in $N \times N$ have cost $1$ and edges in $N \times S$ have cost $2$.
>
> Undirected Case: Any MSF in $G$ consists of a spanning tree in $G[N]$ and one edge from $N \times S$. Importantly, any min-cost spanning tree in $G[N]$ combined with any edge from $N \times S$ forms a MSF. Making use of this property, PW contracts all nodes in $N$ (without any additional computation), and then only needs to compare the number of edges in $N \times \{s\}$ for each $s \in S$ to compute $x_{v,s}$.
>
> Directed Case: This time, any min-cost branching in $G$ consists of an in-tree in $G[N]$ and one edge from $N \times S$. However, since in-trees have dedicated root nodes, we cannot combine any in-tree in $G[N]$ with any edge in $N \times S$. Hence, before contracting the set $N$, Alg. 2 needs to compute the relative number of in-trees rooted at each $v \in N$ in order to calculate $x_{v,s}$ in the next step.
>
> Coming back to the general case, while both algorithms construct a hierarchical structure of subgraphs, the crucial difference is that PW only carries out calculations at the top level of this hierarchy while Alg. 2 needs to carry out calculations at each level. This leads to a blow up of the running time (ignoring log factors) from $\mathcal{O}(n^3)$ (to the best of our understanding) to $\mathcal{O}(n^7)$. That said, our goal, given the broad scope of the paper, was to obtain an exact and poly-time alg. and our upper bound might be improved by a more involved analysis.
> ## Q3
> We agree that, for experts on Markov chains, Thm. 5 is unlikely to be surprising. However, this does not hold for people not being aware of the Markov chain tree theorem, and we expect this to be the case for large parts of the paper's audience. Thus, we think that Sec. 5 is of interest for many readers and by presenting the very short proof, we are upfront about the fact that the result follows easily from known results. That said, we agree weaken our statement to, e.g, "while this result might be surprising for some readers, in fact, it follows rather easily by building upon...".

---

> > ### Comment · Reviewer_rGK4 · 2023-08-15
> >
> > Thank you for addressing my concerns. Discussing the connection to supervised learning earlier in the paper will contribute to its suitability for NeurIps and this will also be a good place to mention that the equivalence of the two rules also follows from results in [1]. Given that, contrary to my initial impression, [1] does not already solve your problem, I will increase my score.

---

### Official Review · Reviewer_vsDd · 2023-07-13

**Soundness:** 4 excellent
**Presentation:** 3 good
**Contribution:** 2 fair
**Rating:** 6
**Confidence:** 3

**Summary:**

This paper studies algorithms for assigning delegation weights in liquid democracy in the setting where participants indicate a set of trusted delegates, along with a ranking describing their preferences over these delegates. Notably, the algorithms it considers permit fractional delegations - i.e., voters have voting weight 1, and they can delegate it fractionally across multiple possible delegates.
The paper considers two algorithms for finding a delegation graph (i.e. an assignment of fractional delegations from voters who wish to delegates, to trusted delegates). These two algorithms are: (1) the random walk rule, a Markov-chain based rule proposed in past work;  and (2) the mixed Borda branching rule, which is a uniform average over all min-cost branchings, called “borda branchings”, where a branching occurs on a weighted digraph representing to which casting voters each voter is willing to delegate to.

The paper makes three main contributions:
(1) Provides a polynomial-time algorithm for computing mixed Borda branching. This algorithm is an adaptation of Fulkerson’s algorithm, and relies on two of its canonical properties (as proven by Fulkerson). This algorithm is of independent interest to the direct power watershed problem.
(2) Proves the equivalence of rules (1) and (2) above.
(3) Shows that rule (1) (and therefore rule (2)) satisfies three axioms: confluence, anonymity, and copy-robustness — a combination of axioms identified in past work that cannot be simultaneously achieved with delegation rules that require voters to delegate their voting weight to a single delegate.


**Strengths:**

- The high-level structure of the paper is clearly laid out

- Some aspects of the paper’s analysis (e.g., axiom iii) are practically motivated, and the setting of liquid democracy in general is well-motivated

- The authors made an effort to make the paper understandable, with multiple diagrams to aid explanations

- The paper speaks to and builds on multiple aspects of the literature (e.g., random walk rule, existing axioms)

- The technical results seem to holds, and there is sufficient technical exposition to understand why the results are true.

**Weaknesses:**

1) The potential impact of this paper is not clear entirely to me. What is the main technical challenge with proving the results and what makes it hard? Why are axioms (i) and (ii) important for a rule to satisfy in practice?

2) I found it very hard to understand several phrases in the introduction. In many cases, it seems that the paper is assuming too much specific knowledge of the liquid democracy literature. For example:
- Line 47: “(i.e., a digraph with a rank function on the edges)” what is a “rank function on edges”?
- Line 51: In the explanation of confluence, what is a “subpath of v1”?
- Line 64: “However, since none of the axioms connects the meaning of the ranks to the decisions of the delegation rule” I do not understand what this sentence means.
- Line 66: “A single top-rank edge to a casting voter should always be chosen over any other delegation path.” I don’t understand what this sentence means.


**Questions:**

Was it previously known whether the random walk rule satisfies these three axioms simultaneously?


**Limitations:**

Yes

---

> ### Author Rebuttal · Authors · 2023-08-09
>
> **Q:** Was it previously known whether the random walk rule satisfies these three axioms simultaneously?\
> **A:** No. For both copy-robustness and confluence, the question whether the random walk rule satisfies (reasonable generalizations of) the axioms were open. In particular, in order to prove the copy-robustness result, we heavily use both the equivalence towards Borda Branching (Theorem 5) as well as the algorithm for computing the Borda Branching outcome (Section 4). Lastly, the fact that the random walk rule satisfies anonymity -- while not explicitly mentioned in the literature -- is rather straightforward and should not come at a surprise.
>
> **Q:** What is the main technical challenge with proving the results and what makes it hard?\
> **A:** The main technical contribution is threefold:  (a) the development of our algorithm and its proof of correctness (Algorithm 2, Theorem 4), (b) the copy-robustness proof (Theorem 7), and (c) the confluence proof (Theorem 8). We describe the main challenges below:
> (a) Two crucial building blocks that serve as the base of the development of the algorithm are two results coming from different research fields: (1) the Markov chain tree theorem, and (2) Fulkerson's algorithm. While the Markov chain tree theorem allows us to count branchings in graphs without costs, the Fulkerson's algorithm provides us with a tool to divide the graph with a cost function into subgraphs. As a result we were able to reduce the task of counting min-cost branchings in the original graph to the task of counting branchings in subgraphs. These subgraphs are not disjoint from on another and the counting has to be done along the hierarchy prescribed by Fulkerson's algorithm, leading to a delicate construction of the subgraphs, including a non-trivial choice of weight functions (not to be confused with cost functions). We also refer to our answer to question 2 of reviewer rGK4 for a detailed comparison between the complexity of our algorithm and the (undirected) Power Watershed.
>
> (b) The main challenge here is that copy-robustnesss is an axiom which is defined across multiple input instances. Hence, we needed to relate the output of our algorithm across different instances. To this end, we proved Lemma 2 (i), which is, to the best of our knowledge, new to our paper.
>
> (c) Here, the main challenge is that confluence prescribes the existence of a probability distribution over walks in the original graph, however, our algorithm for mixed Borda branching contracts the graph. Hence, we build upon a probability distribution over walks in the contracted graph, from which we then derive a distribution over walks in the original graph.
>
> **Q:** Why are axioms (i) and (ii) important for a rule to satisfy in practice?
>
> **A:** (i) Confluence is seen as desirable in order to guarantee the liability of voters for their delegations ([4]). The idea is that voters keep their delegations over time and can therefore evaluate their representative. Consider this simplified situation: If $v$ delegates directly to a set of casting voters and the delegation rule assigns some distribution over these representatives, then we can assume that $v$ takes responsibility for his/her delegation decision (at least over the long run). If now voter $w$ delegates only to $v$, then, confluence prescribes that $w$ receives the same fractional assignment to casting voters as $v$ received. The rationale for that is that $v$ delegates its vote to $w$ and therefore wants his/her voter to be treated as $w$'s vote itself. Otherwise, the liability of $v$ towards its delegation decision is worthless from the point of view of voter $w$. This idea naturally extends to more complicated situations.
>
> (ii) Anonymity is important due to fairness considerations. Consider the simple example given in the introduction of our paper. If $v_1$ would be assigned to $s_1$ (its second preferred delegate) while $v_2$ would be assigned to $s_1$ (via its first preferred delegate), then $v_1$ could complain arguing that the two voters -- despite being in a symmetric situation -- were not treated equally.
>
> **C:** I found it very hard to understand several phrases in the introduction. [...]\
> **A:** We thank the reviewer for the feedback and will revise the introduction for the next version of the paper, in order to make it more accessible for a broader audience. Below, we clarify some of the unclear phrases.
>
> **Q** Line 47: “(i.e., a digraph with a rank function on the edges)” what is a “rank function on edges”?\
> **A** In social choice, the term "ranking" is used interchangeably with the notion of a weak or total order (more precisely, a strict ranking corresponds to a total order and a weak ranking to a weak order). Any (weak or strict) ranking can be naturally represented by a function $r$ mapping the elements to be ranked (in our case edges) to the natural numbers with the interpretation that $e \succeq e'$ if and only if $r(e) \leq r(e')$.
>
> **Q:** Line 51: In the explanation of confluence, what is a “subpath of v1”?\
> **A:** Here, we are considering a path $P$ starting in a voter $v_1$, going via a voter $v_2$, and then ending in some casting voter $s$. The ``remaining subpath of $v_1$'' then refers to the suffix of the path $P$ starting from the occurence of $v_2$ and ending in $s$.
>
> **C:** Line 64: “However, since none of the axioms connects the meaning [...]\
> **A:** Our model assumes that delegations with a lower rank are preferred over delegations with a higher rank (we also use the term costs). However, the axioms (i)-(iii) do not reflect this interpretation. In particular, there exists a delegation rule satisfying the three axioms with the following, undesirable behavior: Consider a trivial instance with one voter that delegates to casting voter $s_1$ with rank 1, and to $s_2$ with rank 2. Then, the rule assigns the voting weight of the voter completely to $s_2$, which would clearly contradict the intention of the voter.

---

> > ### Comment · Reviewer_vsDd · 2023-08-12
> > **Response**
> >
> > I have the read the authors' response and I thank them for the detailed clarifications. I have no further questions.

---

### Author Rebuttal · Authors · 2023-08-09

We would like to express our sincere gratitude to the reviewers for taking the time to review our submission. In the following, we address each of the reviewer's comments and concerns individually.

## References
[1] Fita Sanmartin et al. "Directed Probabilistic Watershed." (2021)\
[2] Couprie et al. "Power watershed: A unifying graph-based optimization framework." (2010)\
[3] Hahn et al. "Probabilistic reachability for parametric Markov models." (2011)\
[4] Brill et al. "Liquid democracy with ranked delegations." (2022)

---

### Decision · Program_Chairs · 2023-09-21

**Decision:**

Accept (spotlight)

**Comment:**

All reviewers are positive and some are excited about the paper. All felt that the paper tackles an important and provides approaches to get around impossibility results in liquid democracy.  This is a great paper for at least the AGT community.

We hope the authors find the reviews helpful. Thanks for submitting to NeurIPS!